# Determining the global citizenship tendencies of international university students for a sustainable world

Adem Uzun[1], Osman Akhan[2]*, Saim Turan[2], Ali Özkaya[3]

1 Department of Gifted Education, Faculty of Education, Cumhuriyet University, Sivas, Turkey,
2 Department of Social Studies Education, Faculty of Education, Akdeniz University, Antalya, Turkey,
3 Department of Mathematics and Science Education, Faculty of Education, Akdeniz University, Antalya, Turkey

* osmanakhan07@gmail.com

## Abstract

This study examines the relationship between the global citizenship awareness of international students studying at universities in Turkey and their orientation toward sustainability. While global citizenship and sustainability have often been explored separately in the literature, the way international students connect these two concepts has not been sufficiently addressed. Therefore, this study aims to fill this gap. A convergent parallel mixed-methods design was employed. Quantitative data were collected from 634 international students (323 women, 311 men) enrolled at four Turkish universities using the Global Citizenship Scale. Qualitative data were gathered through face-to-face interviews with 37 volunteer students. Quantitative findings indicated a moderate level of global citizenship awareness among students. In contrast, qualitative results revealed a more nuanced and higher level of awareness, with students demonstrating the ability to interpret global citizenship from multiple perspectives. Both sets of findings suggest that students develop awareness of environmental and social issues, that active participation enhances their global citizenship awareness, and that they adopt sustainable living practices (e.g., sustainable consumption, resource management). Furthermore, many participants reported applying global citizenship principles in their daily lives. These findings suggest that international students can play a significant role in promoting sustainability and global awareness. The study highlights the importance of integrating global citizenship and sustainability education into higher education curricula to strengthen students' roles as agents of sustainable change.

## Introduction

Since the second half of the 20th century, advances in information and communication technologies have accelerated globalization by increasing economic, political,

**Data availability statement:** The dataset of this study is available upon reasonable request. Data sharing is permitted under the condition that participants' personal information remains confidential. This data access is authorized by the Akdeniz University Social Sciences and Humanities Scientific Research and Publication Ethics Committee (Approval No. 12.08.2025–1291025; document uploaded with this submission). Interested researchers may request access by contacting the committee via email at etikkurul@akdeniz.edu.tr. For follow-up, the committee secretary, Mr. Ali Dikiş, can also be reached at alidikis@akdeniz.edu.tr.

**Funding:** The author(s) received no specific funding for this work.

**Competing interests:** The authors have declared that no competing interests exist.

and cultural interactions among countries. While this process has facilitated the sharing of information and resources, it has also given rise to numerous global challenges, including income inequality, environmental degradation, human rights violations, and forced migration. Addressing these issues, which cannot be resolved within national borders alone, has necessitated the development of new areas of responsibility and models of cooperation. This imperative has, in turn, laid the foundation for the concept of global citizenship, emphasizing that individuals bear responsibility not only to their own societies but to the entire world.

Today, global citizenship embodies a concept shaped by intercultural understanding and shared responsibility, emphasizing active participation in addressing global challenges to create a more just and sustainable world [1]. Global citizens uphold universal principles such as justice, human rights, cultural pluralism, and environmental awareness, recognizing that individuals have responsibilities not only to their own communities but also to the entire world, grounded in shared human values.

Sustainable development is a crucial movement based on the principle that every individual on Earth deserves the best possible living conditions [2]. Achieving this goal requires a collective global effort. When global citizens collaborate with a shared understanding of solutions to these challenges, a more balanced and effective sustainable development process can be achieved. By incorporating a global citizenship perspective, sustainable development can promote a more just, equitable, and environmentally responsible world. Therefore, global citizenship strengthens the social dimension of sustainable development.

The concept of global citizenship must be deeply ingrained in the minds of young people. As the future leaders, decision-makers, and pioneers of social change, it is essential to cultivate a global citizenship mindset in them. When young people are raised with this perspective, they can become agents of positive change not only within their own countries but also on a global scale. In this context, international students have the unique potential to embody global citizenship values and promote sustainability goals by experiencing diverse cultures and developing multifaceted perspectives. An international education experience paves the way for these students to become informed individuals who can contribute meaningfully to both their local and the global community. Consequently, international students who take the initiative to study abroad play a vital role in fostering cultural awareness. To build a more peaceful, just, and sustainable world, these students encourage respect for diverse cultures, traditions, and lifestyles worldwide.

Turkey is a melting pot of diverse cultures. In 2024, it hosted 350,000 international students, making it the country with the largest student population in the European Higher Education Area. Globally, Turkey ranks eighth in terms of international student enrollment. Consequently, international students studying in Turkey have the opportunity to experience a variety of cultures and live in a culturally diverse environment. These experiences enhance students' cultural understanding and global citizenship awareness [3,4].

Previous studies have generally addressed global citizenship awareness within the context of education [5–8] but have not focused on its relationship with sustainability

awareness, especially among international students in Turkey. This study aims to identify the global citizenship tendencies of international university students global citizenship tendencies to examine how these tendencies relate to sustainability awareness. By shedding light on the role of international students in global citizenship and sustainability, this study intends to inform educational policies that promote a sense of global responsibility within higher education institutions. The findings are expected to provide valuable insights into how international students perceive global citizenship and integrate this concept into their daily lives.

## Globalization and the concept of global citizenship

Today, the adjective global is frequently used in a variety of areas, such as global climate change, global epidemics, global economic crises, global trade, global governance, and global security because the sphere, where interaction takes place, has become a place where problems and solutions concern the whole. The concept of globalization, which first emerged in the 1960s, began to be frequently used after the 1980s [9]. By the 1990s, it had become a key word accepted by scientific circles [10].

Globalization is the process of economic, political and cultural integration of the world [11]. Globalization combines economic, social and environmental benefits, and these benefits have transformed, diversified and shaped communities around the world in ways that keep them connected despite being physically distant from each other [12]. The phenomenon of globalization is a multifaceted and complex concept that can be explained socially, politically, technologically and economically [13]. Robertson [14] expresses globalization as the shrinking of the world and at the same time the strengthening of world consciousness.

The multivariate interaction environment created by globalization has redefined the roles of individuals and led them to an understanding of global citizenship. Based on the fact that individuals are part of a community living together in a global world, global citizenship is a concept of citizenship that requires individuals to be sensitive to global issues as well as their responsibilities towards their own countries [15]. In today's interconnected world, global citizenship is not just about awareness; it is also a commitment to promoting positive change across borders [16].

Today, the rapid transformation of local problems into global issues increases the importance of individuals acting with this common consciousness. For example, the COVID-19 pandemic, which emerged in Wuhan, quickly turned into a global health crisis and affected the entire world [17,18]. The Syrian civil war, which began in the 2010s, led to a refugee crisis and triggered the problem of international migration [19]. Again, climate change is making its effects felt worldwide, from the melting of Greenland glaciers to forest fires, and as a problem that threatens global ecosystems, economies, and societies, it affects all societies living on earth [20–23].

To solve these problems for the benefit of humanity and to have a more liveable world, it is necessary to focus not only on the awareness that "comes with global citizenship" but also on how the solutions should be shaped; this can be explained by the concept of sustainability. In today's world where the problems that humanity faces transcend national borders, sustainability is at the center of citizenship [24]. As the challenges we face increasingly transcend national borders, the search for sustainability emerges as a vital framework to promote global cooperation and ensure a livable future for everyone [25]. Thus, in its 2030 agenda, the United Nations aims to develop intercultural understanding, tolerance, mutual respect and an ethic of shared responsibility through global citizenship; by acknowledging the natural and cultural diversity of the world, it has stated that "all cultures and civilizations can contribute to sustainable development and are an important supporter of this" [26].

## The relationship between sustainability and global citizenship

Sustainability, fundamentally grounded in opposing monopolization and individualization, essentially refers to the state or quality of something that can be maintained and continued over time. Consequently, sustainability has evolved beyond being merely a modern environmental movement; it has become a mindset and mode of action that emphasizes the

 

cultures of societies. As a way of thinking, sustainability embodies an approach aimed at bridging differences between societies and living together in a desirable future [27].

Sustainable development, on the other hand, addressing the cultures of all societies and acting according to their specific needs, as it is fundamentally a social activity plan. Simultaneously, sustainable development aims to provide humanity with better living conditions and enhance overall welfare. It has become a central field of study because it encompasses political concepts such as freedom, social justice, rights, and law [28]. Furthermore, sustainable development has emerged as a set of universal goals designed to meet the needs of the present generation while considering those of future generations. It also focuses on combating inequalities, recognizing the importance of cooperation, and supporting societies [29]. These objectives are encapsulated in the Sustainable Development Goals, which consist of 17 main goals and 169 sub-goals [30].

The Sustainable Development Goals explain the concept of global citizenship, particularly within the framework of Goal 16: Peace, Justice, and Strong Institutions. This goal aims to promote justice, peace, equality, human rights, and robust institutions. Additionally, Goal 4: Quality Education, seeks to advance the idea of global citizenship by encouraging individuals to become informed about global issues, respect diverse cultures, and engage as active, responsible members of the global community through education. Individuals who are sensitive to global challenges, capable of communicating across cultures, and possess a multifaceted perspective can emerge as leaders capable of transforming societies. In this regard, international students hold significant potential.

Studying abroad and engaging with diverse cultures can broaden individuals' perspectives, enabling them to view the world in new and varied ways. Participation in international educational mobility programs enhances awareness of our highly interconnected world and highlights the role everyone plays in promoting sustainable development [31]. These programs immerse higher education students in a learning process where they acquire fundamental knowledge, skills, and attitudes critical to achieving the Sustainable Development Goals [32]. Schnepf and Colagrossi [33] argue that expanding student mobility via the Erasmus program serves as a valuable policy tool for fostering intercultural understanding and enhancing employability in the globalized labor market. In times of conflict, exchange programs offer young people opportunities to build friendships with peers from different parts of the world. Moreover, the increasingly globalized economy demands that the future workforce possess the skills necessary to interact effectively with individuals from other cultures [34].

Developing global citizenship can be fostered through educational experiences gained by living abroad [35]. Students who study abroad are reported to be more willing to change their perspectives on global issues and benefit from their international experiences by enhancing their ability to adapt to new situations and communicate effectively [36]. Gibson, Rimmington, and Landwehr-Brown [37] stated that successful intercultural collaboration and effective communication among people from different cultures can cultivate a greater sense of global awareness.

Since the problems the world faces today—from climate change to social inequality—require solutions that transcend national borders, it is important to understand international university students' perspectives on global citizenship through various tools. Indeed, as international students engage with different cultures, they gain valuable experience in issues such as global citizenship, intercultural interaction, and the impact of local problems on people and societies worldwide. Indeed, international students who develop an understanding of sustainability that they will carry into their lives through their education and interactions become important actors for a sustainable future with the different perspectives and experiences they acquire [38].

## Method

### Research model

This study used a mixed-methods approach, incorporating both quantitative and qualitative data collection and analysis. Quantitative and qualitative methods each provide distinct types of data and capture different facets of the phenomenon under investigation. By integrating these findings, the study's scope, depth, detail, and validity are enhanced [39].

Mixed-method designs are often preferred in educational and social science research when comprehensive, in-depth information is required, allowing for the combined use of quantitative and qualitative data. In this study, quantitative and qualitative data were collected simultaneously and completed in a single phase. Accordingly, the research was designed as a convergent parallel mixed-methods design [40]. In this design, quantitative and qualitative data are collected concurrently, analyzed separately, and then integrated and interpreted together [41].

## Study group

The study group for this research consists of 634 international students (323 females, 311 males) enrolled at four different universities in Turkey. Among these students, 60 reported coming from Asia, 154 from Europe, 118 from Russia/Eurasia, 103 from Sub-Saharan Africa, and 199 from the Middle East and North Africa region. Participants were contacted by obtaining their phone numbers and email addresses through the international relations offices of their respective universities.

The purposive sampling method was employed to select the study group. When choosing the students, the criteria included having completed at least one semester of education in Turkey as international students and possessing English language proficiency. To meet these requirements, universities located in the cities where the authors work were selected. Students were contacted via email in English and asked to confirm their English proficiency. The email also informed them that the interviews would be conducted in English. Only students who agreed to all the terms and responded to the invitation email were included in the study group.

The Global Citizenship Scale, which represents the quantitative component of the research, was administered via a Google Form. At the end of the form, participants were asked, "Would you like to contribute to this study through a face-to-face interview?" Forty-three students volunteered by responding and agreeing to participate. Students were reminded to complete the scale, sent through the Google Form, prior to attending the interview. Six students were unable to participate for various reasons (e.g., illness, withdrawal), leaving 37 students who comprised the study group for the qualitative phase of the research.

## The data collection stage and ethical procedure

Data for this study were collected using a quantitative scale and qualitative interviews. Data collection took place from October 15 to December 25, 2024. The study was approved by Akdeniz University's Social Sciences and Humanities Scientific Research and Publication Ethics Committee.

In the quantitative phase of the study, after the necessary permissions were obtained, the "Global Citizen Scale" developed by Reysen and Katzarska-Miller [42] was used. The scale consists of 22 items across nine factors and uses a seven-point Likert-type response format. Items are rated from "Strongly Disagree (1)" to "Strongly Agree (7)," allowing for a total score ranging from 22 to 154 points. The scale includes the following factors: Normative Environment (items 1–4); Global Awareness (items 5–8); Global Citizenship Identification (items 9 and 10); Intergroup Empathy (items 11 and 12); Valuing Diversity (items 13 and 14); Social Justice (items 15 and 16); Environmental Sustainability (items 17 and 18); Intergroup Helping (items 19 and 20); and Responsibility to Act (items 21 and 22). The internal consistency coefficients (Cronbach's alpha) for these subfactors are as follows: Normative Environment (.82), Global Awareness (.80), Global Citizenship Identification (.89), Intergroup Empathy (.76), Valuing Diversity (.91), Social Justice (.74), Environmental Sustainability (.76), Intergroup Helping (.76), and Responsibility to Act (.78).

For the qualitative component of the study, nine interview questions were developed. These questions were initially designed to directly capture participants' perceptions related to the Global Citizen Scale, which was developed by Reysen and Katzarska-Miller [42]. After drafting the interview questions, expert feedback was obtained from two faculty members with expertise in global citizenship. Based on their input, the interview form was refined and then pilot-tested with five international university students who were not part of the study sample. Points of confusion identified during this pilot

 

phase were addressed, and the interview form was revised accordingly. The finalized interview form was then administered through face-to-face interviews with 37 international university students selected as the study group. The semi-structured interviews were conducted in person, audio-recorded, and subsequently transcribed.

## Analysis of data

The quantitative dataset was analyzed using SPSS version 22, while Jamovi [43]was employed for assumption checks and additional analyses. Normality was assessed through skewness–kurtosis measures and the Shapiro–Wilk test, following the tolerance ranges reported in [44,45]. Normality was confirmed only for the 18–20 age subgroup concerning the ideal environment, global awareness, and total scores (p > .05); all other scores violated normality assumptions (p < .05). Consequently, nonparametric tests were applied: Spearman's correlation for associations, Mann–Whitney U tests for two-group comparisons (with effect size reported as biserial correlation [46]), and Kruskal–Wallis tests for comparisons involving multiple groups (with effect size reported as epsilon squared). For the Kruskal–Wallis test, epsilon squared was interpreted as follows: 0.01 indicates a small effect, 0.09 a medium effect, and 0.25 or higher a large effect [47].

The qualitative dataset from the research was imported into the MAXQDA 2020 Plus qualitative data analysis software and analyzed using the thematic analysis method. This software offers researchers a systematic approach to managing large volumes of qualitative data, facilitating comparisons between categories and the extraction of themes from the data [48].

In the analysis of the study's qualitative data, all authors collaboratively coded the data and continued the analysis until consensus was reached in the coding and theme development processes. The interview data were reviewed by two independent researchers prior to coding to ensure internal consistency in code interpretation and to minimize bias [49]. Additionally, the constant comparison method was employed during coding to enhance the accuracy of the themes by systematically comparing similarities and differences within the data. Finally, in the findings section, the results were supported by direct quotes from participants, identified using abbreviations such as F1 (Female 1) and M2 (Male 2).

## Findings

### Findings on the analysis of quantitative data

#### Findings regarding the global citizenship scale and its sub-dimensions

When Table 1 is examined, it is observed that individuals for the normative environment sub-dimension of the global citizenship scale are $min = 4.00; maks = 28.00; \overline{X} = 17.82 \pm 6.01$.

**Table 1. Descriptive statistics for global citizenship total score and its sub-dimensions.**

| Factors | Min. | Max. | Mean | Sd | Skewness | Kurtosis | Shapiro-Wilk |
|---|---|---|---|---|---|---|---|
| Normative Environment | 4.00 | 28.00 | 17.82 | 6.01 | −.353 | −.364 | .000 |
| Global Awareness | 4.00 | 28.00 | 18.95 | 5.90 | −.468 | −.273 | .000 |
| Global Citizenship Identification | 2.00 | 14.00 | 9.07 | 3.35 | −.292 | −.605 | .000 |
| Intergroup Empathy | 2.00 | 14.00 | 9.80 | 3.29 | −.582 | −.416 | .000 |
| Valuing Diversity | 2.00 | 14.00 | 10.37 | 3.26 | −.801 | −.048 | .000 |
| Social Justice | 2.00 | 14.00 | 10.88 | 3.37 | −1.081 | .302 | .000 |
| Environmental Sustainability | 2.00 | 14.00 | 10.77 | 3.30 | −1.036 | .326 | .000 |
| Intergroup Helping | 2.00 | 14.00 | 10.51 | 3.43 | −.834 | −.195 | .000 |
| Responsibility to Act | 2.00 | 14.00 | 9.98 | 3.25 | −.749 | −.090 | .000 |
| Global Citizenship Total Score | 22.00 | 154.00 | 108.20 | 28.23 | −.824 | .542 | .000 |

For the global awareness sub-dimension, it was observed that individuals had $min = 4.00; maks = 28.00; \overline{X} = 18.95 \pm 5.90$. For the global citizenship identification sub-dimension, it was observed that individuals had $min = 2.00; maks = 14.00; \overline{X} = 9.07 \pm 3.35$. For the intergroup empathy sub-dimension, it was observed that individuals had $min = 2.00; maks = 14.00; \overline{X} = 9.80 \pm 3.29$. For the valuing diversity sub-dimension, it was observed that individuals had $min = 2.00; maks = 14.00; \overline{X} = 10.37 \pm 3.26$. For the Social Justice sub-dimension, it was observed that individuals had $min = 2.00; maks = 14.00; \overline{X} = 10.88 \pm 3.37$. For the environmental sustainability sub-dimension, it was observed that individuals had $min = 2.00; maks = 14.00; \overline{X} = 10.77 \pm 3.30$. For the intergroup helping sub-dimension, it was observed that individuals had $min = 2.00; max = 14.00; \overline{X} = 10.51 \pm 3.43$. For the responsibility to act sub-dimension, it was observed that individuals had $min = 2.00; maks = 14.00; \overline{X} = 9.98 \pm 3.25$. For the global citizenship total score, it was observed that individuals had $min = 22.00; maks = 154.00; \overline{X} = 108.20 \pm 28.23$. The total score and sub-dimensions of the scale do not show normal distribution for demographic variable groups ($p < 0.05$). Due to this situation, non-parametric analysis methods were used for relationship and difference tests. In this context, Mann Whitney U test was applied for demographic variables with two groups and Kruskall Wallis H test was applied for variables with more than two groups to examine the difference between demographic variable groups. Spearman Brown correlation test was used for relationship analysis.

When Table 2 is examined, cut-off scores were determined on an item basis in order to determine the levels of the Global Citizenship Scale. When determining the cut-off scores, five categories were determined to determine the global citizenship levels of the group according to the 7-point Likert scale. Then, the average scores for the sub-dimensions and the main scale dimension were divided by the number of items to determine the levels. Accordingly,

1.00-2.49 is at a very low level

2.50-3.99 is at a low level

4.00-5.49 is at a medium level

5.50-6.49 is at a high level

6.50-7.00 is at a very high level

When the mean scores of the normative environment dimension of the global citizenship scale are examined, it is seen that the group is a citizen of the normative environment at a moderate level ($\overline{X} = 4.45$). When the average of the global awareness dimension scores is examined, it is seen that the group has a medium level of global awareness ($\overline{X} = 4.73$). When the global citizenship identification dimension average is examined, it is seen that the group has a medium level of global citizenship identification ($\overline{X} = 4.53$). When the intergroup empathy dimension average was examined, it was seen that the group had a medium level of intergroup empathy ($\overline{X} = 4.90$). When the mean of the valuation diversity dimension was examined, it was seen that the group had a medium level of valuation diversity ($\overline{X} = 5.18$). When the average of the social justice dimension was examined, it was seen that the group had a moderate level of social justice ($\overline{X} = 5.44$). When the environmental sustainability dimension average was examined, it was seen that the group had a medium level of environmental sustainability ($\overline{X} = 5.38$). When the mean of the intergroup helping dimension was examined, it was seen that the group had a medium level of intergroup helping ($\overline{X} = 5.25$). When the mean of the responsibility to act dimension was examined, it was seen that the group had a medium level of responsibility to act ($\overline{X} = 4.99$). When the average of the global citizenship scale scores was examined, it was observed that the group was a moderate level ($\overline{X} = 4.91$) of global citizens.

Table 2. Scoring system used in the global citizenship scale.

|  | Very Low | Low | Medium | High | Very High |
|---|---|---|---|---|---|
| Items | 1.00-2.49 | 2.50-3.99 | 4.00-5.49 | 5.50-6.49 | 6.50-7.00 |

## Findings regarding the regional variable

When Table 3 is examined, it is seen that there is no statistically significant difference between the normative environment, global awareness, global citizenship identification, intergroup empathy, valuing diversity, social justice, environmental sustainability, intergroup helping, responsibility to act and global citizenship total score means according to the country variable; $X^2$=7.13, p=.129; $X^2$=1.24, p=.871; $X^2$=3.24, p=.519; $X^2$=2.98, p=.561; $X^2$=3.22, p=.522; $X^2$=5.94, p=.204; $X^2$=8.14, p=.086; $X^2$=5.98, p=.201. It was observed that there was a statistically significant difference between the global awareness and environmental sustainability score averages according to the country variable; $X^2$= 18.89, p=.001; $X^2$=10.65, p=.031. Dwass-Steel-Critchlow-Fligner pairwise comparison test was performed to determine which groups the significant difference was. As a result of the pairwise comparisons, it was seen that the mean score of Europe for the global awareness sub-dimension was significantly higher than the mean score of Russia/Eurasia and the Middle East/North Africa; and the mean score of Europe for the environmental sustainability sub-dimension was significantly higher than the mean score of Russia/Eurasia. When the effect size of the difference in question was calculated, it could be said that it was small.

When the global citizenship scale total score averages are examined according to geographical regions, the European group ($X_{mean} = 112, 40$) is the highest, the Asian group ($X_{mean} = 109, 37$) is second, the Sub-Saharan African group ($X_{mean} = 109, 33$) is third, the Middle East/North African group ($X_{mean} = 106, 18$) is fourth and the Russian/Eurasian group ($X_{mean} = 104, 56$) is last.

## Findings regarding the social media variable

The frequencies of the social media tools used by the participants are given in Table 4.

It was observed that the most frequently used social media tools by the participants were Instagram (510) and Facebook (398); the least frequently used social media tools were Reddit (43) and Discord (80). The correlation coefficient for the social media score and scale score was calculated and is given in Table 5.

It was observed that there was no statistically significant relationship between the social media score and normative environment, global awareness, intergroup empathy, valuing diversity, social justice, environmental sustainability, intergroup helping, responsibility to act and global citizenship total score, p>0.05; and there was a statistically significant positive and small-level relationship with global citizenship identification, p<0.05.

## Findings regarding the gender variable

When Table 6 is examined, it is seen that there is no statistically significant difference between the normative environment, global awareness, global citizenship identification, valuing diversity, social justice, environmental sustainability, intergroup helping, responsibility to act, global citizenship total score averages according to the gender variable; U=46665, p=.122; U=48593, p=.478; U=47902, p=.310; U=49153, p=.638; U=48067, p=.340; U=48368, p=.413; U=49111, p=.623; U=48099, p=.353; U=50186, p=.986.

It was observed that there was a statistically significant difference between the intergroup empathy scores according to the gender variable, U=45729, p=.049. The intergroup empathy mean score of men is significantly higher than the mean score of women. When the effect size of the difference in question is calculated, it can be said that it is small.

## Findings regarding the global event participation variable

When Table 7 is examined, it is seen that there is a statistically significant difference between the averages of normative environment, global awareness, global citizenship identification, intergroup empathy, valuing diversity, social justice, environmental sustainability, intergroup helping, responsibility to act and global citizenship total score according to the variable of participation in global events; U=26064, p=.001; U=23684, p=.001; U=23654 p=.001; U=26111 p=.001; U=23582 p=.001; U=27335 p=.001; U=27793 p=.001; U=25500 p=.001; U=25339 p=.001; U=22854 p=.001. The

**Table 3. Kruskal Wallis H test on country variable for global citizenship scale.**

| Variable | Geographical Region | N | $X_{mean}$ | $S_x$ | $X^2$ | sd | p | Difference | $\varepsilon^2$ |
|---|---|---|---|---|---|---|---|---|---|
| Normative Environment | Asia | 60 | 18.52 | 6.16 | 7.13 | 4 | .129 | | |
| | Europe | 154 | 18.79 | 5.40 | | | | | |
| | Russia/Eurasia | 118 | 17.20 | 6.04 | | | | | |
| | Sub-Saharan Africa | 103 | 17.77 | 6.89 | | | | | |
| | Middle East/North Africa | 199 | 17.27 | 5.88 | | | | | |
| Global Awareness | Asia | 60 | 18.58 | 6.03 | 18.89 | 4 | .001 | 2 > 3 | .030 |
| | Europe | 154 | 20.52 | 5.08 | | | | 2 > 5 | |
| | Russia/Eurasia | 118 | 18.47 | 6.08 | | | | | |
| | Sub-Saharan Africa | 103 | 19.28 | 6.57 | | | | | |
| | Middle East/North Africa | 199 | 17.98 | 5.78 | | | | | |
| Global Citizenship Identification | Asia | 60 | 9.32 | 3.51 | 1.24 | 4 | .871 | | |
| | Europe | 154 | 9.01 | 3.20 | | | | | |
| | Russia/Eurasia | 118 | 9.10 | 3.32 | | | | | |
| | Sub-Saharan Africa | 103 | 9.28 | 3.48 | | | | | |
| | Middle East/North Africa | 199 | 8.93 | 3.39 | | | | | |
| Intergroup Empathy | Asia | 60 | 9.68 | 3.20 | 3.24 | 4 | .519 | | |
| | Europe | 154 | 10.23 | 3.03 | | | | | |
| | Russia/Eurasia | 118 | 9.56 | 3.49 | | | | | |
| | Sub-Saharan Africa | 103 | 9.60 | 3.45 | | | | | |
| | Middle East/North Africa | 199 | 9.76 | 3.33 | | | | | |
| Valuing Diversity | Asia | 60 | 10.85 | 3.31 | 2.98 | 4 | .561 | | |
| | Europe | 154 | 10.45 | 2.81 | | | | | |
| | Russia/Eurasia | 118 | 10.07 | 3.39 | | | | | |
| | Sub-Saharan Africa | 103 | 10.45 | 3.30 | | | | | |
| | Middle East/North Africa | 199 | 10.30 | 3.48 | | | | | |
| Social Justice | Asia | 60 | 10.83 | 3.36 | 3.22 | 4 | .522 | | |
| | Europe | 154 | 11.36 | 2.84 | | | | | |
| | Russia/Eurasia | 118 | 10.44 | 3.55 | | | | | |
| | Sub-Saharan Africa | 103 | 10.77 | 3.62 | | | | | |
| | Middle East/North Africa | 199 | 10.87 | 3.53 | | | | | |
| Environmental Sustainability | Asia | 60 | 10.73 | 3.17 | 10.65 | 4 | .031 | 2 > 3 | .017 |
| | Europe | 154 | 11.44 | 2.74 | | | | | |
| | Russia/Eurasia | 118 | 10.10 | 3.43 | | | | | |
| | Sub-Saharan Africa | 103 | 10.96 | 3.32 | | | | | |
| | Middle East/North Africa | 199 | 10.58 | 3.58 | | | | | |
| Intergroup Helping | Asia | 60 | 10.58 | 3.42 | 5.94 | 4 | .204 | | |
| | Europe | 154 | 10.55 | 2.99 | | | | | |
| | Russia/Eurasia | 118 | 10.13 | 3.23 | | | | | |
| | Sub-Saharan Africa | 103 | 10.80 | 3.54 | | | | | |
| | Middle East/North Africa | 199 | 10.56 | 3.81 | | | | | |
| Responsibility to Act | Asia | 60 | 10.27 | 2.97 | 8.14 | 4 | .086 | | |
| | Europe | 154 | 10.05 | 3.10 | | | | | |
| | Russia/Eurasia | 118 | 9.48 | 3.02 | | | | | |
| | Sub-Saharan Africa | 103 | 10.43 | 3.45 | | | | | |
| | Middle East/North Africa | 199 | 9.91 | 3.47 | | | | | |

*(Continued)*

**Table 3.** (Continued)

| Variable | Geographical Region | N | $X_{mean}$ | $S_x$ | $X^2$ | sd | p | Difference | $\varepsilon^2$ |
|---|---|---|---|---|---|---|---|---|---|
| Global Citizenship Total Score | Asia | 60 | 109.37 | 29.46 | 5.98 | 4 | .201 | | |
| | Europe | 154 | 112.40 | 24.09 | | | | | |
| | Russia/Eurasia | 118 | 104.56 | 27.59 | | | | | |
| | Sub-Saharan Africa | 103 | 109.33 | 30.78 | | | | | |
| | Middle East/North Africa | 199 | 106.18 | 29.60 | | | | | |

**Table 4.** Frequency values of social media tools used.

| Social Media Tool | N | % |
|---|---|---|
| Discord | 80 | 12.62 |
| Facebook | 398 | 62.78 |
| Instagram | 510 | 80.44 |
| LinkedIn | 173 | 27.29 |
| Pinterest | 150 | 23.66 |
| Reddit | 43 | 6.78 |
| TikTok | 305 | 48.11 |
| Twitter (now X) | 271 | 42.74 |

**Table 5.** Spearman Brown test for social media score for global citizenship scale.

| N = 634 | Social Media Score | P Value |
|---|---|---|
| Normative Environment | .024 | .552 |
| Global Awareness | .072 | .072 |
| Global Citizenship Identification | .111 | .005 |
| Intergroup Empathy | .066 | .096 |
| Valuing Diversity | .004 | .924 |
| Social Justice | .013 | .750 |
| Environmental Sustainability | −.014 | .717 |
| Intergroup Helping | .039 | .327 |
| Responsibility to Act | .049 | .218 |
| Global Citizenship Total Score | .059 | .140 |

sub-dimension and total score averages of the participants who participated in a global event are significantly higher than the score averages of the participants who did not participate. When the effect size for the difference in question is calculated, it can be said that it is medium for the total score and small for the sub-dimensions.

### Findings regarding the variable of being a member of a global association

When Table 8 is examined, it is seen that there is no statistically significant difference between the mean scores of intergroup empathy, social justice and environmental sustainability according to the variable of being a member of a global association; U = 24204, p = .082; U = 23920, p = .053; U = 25312, p = .275. According to the variable of being a member of a global association, there is a statistically significant difference between the averages of normative environment, global awareness, global citizenship identification, valuing diversity, intergroup helping, responsibility to act and global citizenship total score.; U = 22580, p = .007; U = 21912, p = .002; U = 21917, p = .002; U = 22499, p = .006; U = 21711, p = .001; U = 22875,

**Table 6. Findings regarding the Mann Whitney U test for the gender variable for global citizenship scale.**

| Variable | Gender | N | $X_{mean}$ | Median | $S_x$ | U | p | Binary Correlation |
|---|---|---|---|---|---|---|---|---|
| Normative Environment | Male | 311 | 17.38 | 18.00 | 6.14 | 46665 | .122 | |
| | Female | 323 | 18.26 | 19.00 | 5.88 | | | |
| Global Awareness | Male | 311 | 19.04 | 20.00 | 6.14 | 48593 | .478 | |
| | Female | 323 | 18.88 | 19.00 | 5.67 | | | |
| Global Citizenship Identification | Male | 311 | 8.89 | 9.00 | 3.50 | 47902 | .310 | |
| | Female | 323 | 9.26 | 9.00 | 3.20 | | | |
| Intergroup Empathy | Male | 311 | 10.04 | 10.00 | 3.28 | 45729 | .049 | .090 |
| | Female | 323 | 9.57 | 10.00 | 3.30 | | | |
| Valuing Diversity | Male | 311 | 10.23 | 11.00 | 3.50 | 49153 | .638 | |
| | Female | 323 | 10.51 | 11.00 | 3.02 | | | |
| Social Justice | Male | 311 | 10.72 | 12.00 | 3.53 | 48067 | .340 | |
| | Female | 323 | 11.06 | 12.00 | 3.22 | | | |
| Environmental Sustainability | Male | 311 | 10.77 | 12.00 | 3.50 | 48368 | .413 | |
| | Female | 323 | 10.79 | 12.00 | 3.11 | | | |
| Intergroup Helping | Male | 311 | 10.49 | 12.00 | 3.63 | 49111 | .623 | |
| | Female | 323 | 10.54 | 11.00 | 3.24 | | | |
| Responsibility to Act | Male | 311 | 9.80 | 10.00 | 3.43 | 48099 | .353 | |
| | Female | 323 | 10.15 | 11.00 | 3.08 | | | |
| Global Citizenship Total Score | Male | 311 | 107.35 | 113.00 | 30.34 | 50186 | .986 | |
| | Female | 323 | 109.02 | 113.00 | 26.06 | | | |

**Table 7. Findings regarding the Mann Whitney U test for the global event participation variable for global citizenship scale.**

| Variable | Participation in Global Event | N | $X_{mean}$ | Median | $S_x$ | U | p | Binary Correlation |
|---|---|---|---|---|---|---|---|---|
| Normative Environment | No | 503 | 17.44 | 17.00 | 5.98 | 26064 | .001 | .209 |
| | Yes | 131 | 19.3 | 20.0 | 5.95 | | | |
| Global Awareness | No | 503 | 18.41 | 18.00 | 5.83 | 23684 | .001 | .281 |
| | Yes | 131 | 21.1 | 22.0 | 5.71 | | | |
| Global Citizenship Identification | No | 503 | 8.75 | 9.00 | 3.34 | 23654 | .001 | .282 |
| | Yes | 131 | 10.4 | 10.0 | 3.07 | | | |
| Intergroup Empathy | No | 503 | 9.58 | 10.00 | 3.31 | 26111 | .001 | .207 |
| | Yes | 131 | 10.7 | 11.0 | 3.13 | | | |
| Valuing Diversity | No | 503 | 10.10 | 10.00 | 3.21 | 23582 | .001 | .284 |
| | Yes | 131 | 11.4 | 13.0 | 3.25 | | | |
| Social Justice | No | 503 | 10.70 | 12.00 | 3.42 | 27335 | .002 | .170 |
| | Yes | 131 | 11.6 | 13.0 | 3.12 | | | |
| Environmental Sustainability | No | 503 | 10.59 | 11.00 | 3.37 | 27793 | .005 | .156 |
| | Yes | 131 | 11.5 | 12.0 | 2.97 | | | |
| Intergroup Helping | No | 503 | 10.27 | 11.00 | 3.44 | 25500 | .001 | .226 |
| | Yes | 131 | 11.5 | 13.0 | 3.23 | | | |
| Responsibility to Act | No | 503 | 9.75 | 10.00 | 3.23 | 25339 | .001 | .231 |
| | Yes | 131 | 10.9 | 12.0 | 3.22 | | | |
| Global Citizenship Total Score | No | 503 | 105.59 | 110.00 | 27.83 | 22854 | .001 | .306 |
| | Yes | 131 | 118.2 | 125.0 | 27.61 | | | |

Table 8. Findings regarding the Mann Whitney U test for the variable of being a member of a global association for global citizenship scale.

| Variable | Membership Status | N | $X_{mean}$ | Median | $S_x$ | U | p | Biserial Correlation |
|---|---|---|---|---|---|---|---|---|
| Normative Environment | No | 532 | 17.59 | 18.00 | 5.94 | 22580 | .007 | .168 |
| | Yes | 102 | 19.08 | 20.0 | 6.32 | | | |
| Global Awareness | No | 532 | 18.72 | 19.00 | 5.66 | 21912 | .002 | .192 |
| | Yes | 102 | 20.20 | 22.0 | 6.91 | | | |
| Global Citizenship Identification | No | 532 | 8.92 | 9.00 | 3.29 | 21917 | .002 | .192 |
| | Yes | 102 | 9.89 | 10.0 | 3.58 | | | |
| Intergroup Empathy | No | 532 | 9.73 | 10.00 | 3.26 | 24204 | .082 | |
| | Yes | 102 | 10.20 | 11.0 | 3.49 | | | |
| Valuing Diversity | No | 532 | 10.26 | 11.00 | 3.20 | 22499 | .006 | .171 |
| | Yes | 102 | 10.97 | 12.0 | 3.51 | | | |
| Social Justice | No | 532 | 10.83 | 12.00 | 3.32 | 23920 | .053 | |
| | Yes | 102 | 11.19 | 13.0 | 3.69 | | | |
| Environmental Sustainability | No | 532 | 10.75 | 11.00 | 3.24 | 25312 | .275 | |
| | Yes | 102 | 10.91 | 12.0 | 3.63 | | | |
| Intergroup Helping | No | 532 | 10.39 | 11.00 | 3.36 | 21711 | .001 | .200 |
| | Yes | 102 | 11.20 | 13.0 | 3.72 | | | |
| Responsibility to Act | No | 532 | 9.90 | 10.00 | 3.15 | 22875 | .011 | .157 |
| | Yes | 102 | 10.42 | 12.0 | 3.78 | | | |
| Global Citizenship Total Score | No | 532 | 107.08 | 112.00 | 26.75 | 20938 | .001 | .228 |
| | Yes | 102 | 114.05 | 128.0 | 34.52 | | | |

p = .011; U = 20938, p = .001. It was observed that the mean sub-dimension scores of the participants who were members of the association were significantly higher than those who were not members of the association. When the effect size for this difference was calculated, it could be said that it was small.

**Findings regarding age variable**

When Table 9 is examined, it is seen that there is no statistically significant difference between the normative environment, global awareness and global citizenship identification mean scores according to the age variable.; $X^2$ = 4.64, p = .200; $X^2$ = 3.85, p = .278; $X^2$ = 6.40, p = .094. It was observed that there was a statistically significant difference between the averages of intergroup empathy, valuing diversity, social justice, environmental sustainability, intergroup helping, responsibility to act and global citizenship total score according to the age variable; $X^2$ = 24.41, p = .001; $X^2$ = 16.61, p = .001; $X^2$ = 19.93, p = .001; $X^2$ = 16.30, p = .001; $X^2$ = 28.93, p = .001; $X^2$ = 20.45, p = .001; $X^2$ = 17.52, p = .001. The Dwass-Steel-Critchlow-Fligner pairwise comparison test was conducted to determine between which groups the significant difference occurred. As a result of the pairwise comparisons, it was observed that the sub-dimension scores of participants over the age of 27 were significantly higher than those of other age groups. When the effect size for the difference in question was calculated, it could be said that it was small.

**Findings on the analysis of qualitative data**

**Findings regarding students' definitions of global citizenship**

In the interview conducted with the students in the study group, they were asked how they defined global citizenship. The students' responses are shown in Fig 1:

PLOS One | https://doi.org/10.1371/journal.pone.0335817   October 31, 2025
12 / 28

**Table 9. Kruskal Wallis H test for age variable for global citizenship scale.**

| Variable | Age | N | $X_{ort}$ | $S_x$ | $X^2$ | sd | p | Difference | $\varepsilon^2$ |
|---|---|---|---|---|---|---|---|---|---|
| Normative Environment | 18-20 | 83 | 17.04 | 5.49 | 4.64 | 3 | .200 | | |
| | 21-23 | 228 | 17.95 | 5.79 | | | | | |
| | 24-26 | 125 | 17.36 | 6.28 | | | | | |
| | 27+ | 198 | 18.31 | 6.31 | | | | | |
| Global Awareness | 18-20 | 83 | 18.49 | 5.66 | 3.85 | 3 | .278 | | |
| | 21-23 | 228 | 19.05 | 5.43 | | | | | |
| | 24-26 | 125 | 18.28 | 6.20 | | | | | |
| | 27+ | 198 | 19.48 | 6.30 | | | | | |
| Global Citizenship Identification | 18-20 | 83 | 8.67 | 2.99 | 6.40 | 3 | .094 | | |
| | 21-23 | 228 | 8.86 | 3.31 | | | | | |
| | 24-26 | 125 | 9.09 | 3.51 | | | | | |
| | 27+ | 198 | 9.49 | 3.42 | | | | | |
| Intergroup Empathy | 18-20 | 83 | 9.61 | 2.97 | 24.41 | 3 | .001 | 1<4 | .039 |
| | 21-23 | 228 | 9.19 | 3.35 | | | | 2<4 | |
| | 24-26 | 125 | 9.62 | 3.62 | | | | | |
| | 27+ | 198 | 10.70 | 2.96 | | | | | |
| Valuing Diversity | 18-20 | 83 | 10.39 | 2.75 | 16.61 | 3 | .001 | 2<4 | .026 |
| | 21-23 | 228 | 9.94 | 3.35 | | | | 3<4 | |
| | 24-26 | 125 | 10.04 | 3.47 | | | | | |
| | 27+ | 198 | 11.07 | 3.13 | | | | | |
| Social Justice | 18-20 | 83 | 10.51 | 2.94 | 19.93 | 3 | .001 | 1<4 | .031 |
| | 21-23 | 228 | 10.75 | 3.55 | | | | 2<4 | |
| | 24-26 | 125 | 10.22 | 3.66 | | | | 3<4 | |
| | 27+ | 198 | 11.64 | 3.04 | | | | | |
| Environmental Sustainability | 18-20 | 83 | 10.33 | 2.84 | 16.30 | 3 | .001 | 1<4 | .026 |
| | 21-23 | 228 | 10.76 | 3.44 | | | | 3<4 | |
| | 24-26 | 125 | 10.25 | 3.41 | | | | | |
| | 27+ | 198 | 11.32 | 3.20 | | | | | |
| Intergroup Helping | 18-20 | 83 | 10.02 | 2.94 | 28.93 | 3 | .001 | 1<4 | .046 |
| | 21-23 | 228 | 10.15 | 3.59 | | | | 2<4 | |
| | 24-26 | 125 | 9.97 | 3.58 | | | | 3<4 | |
| | 27+ | 198 | 11.48 | 3.16 | | | | | |
| Responsibility to Act | 18-20 | 83 | 9.55 | 2.76 | 20.45 | 3 | .001 | 1<4 | .032 |
| | 21-23 | 228 | 9.77 | 3.22 | | | | 2<4 | |
| | 24-26 | 125 | 9.50 | 3.48 | | | | 3<4 | |
| | 27+ | 198 | 10.70 | 3.25 | | | | | |
| Global Citizenship Total Score | 18-20 | 83 | 104.61 | 23.56 | 17.52 | 3 | .001 | 1<4 | .028 |
| | 21-23 | 228 | 106.41 | 27.42 | | | | 2<4 | |
| | 24-26 | 125 | 104.34 | 31.95 | | | | 3<4 | |
| | 27+ | 198 | 114.20 | 27.68 | | | | | |

When the responses of the students regarding their definitions of global citizenship are examined, it is seen that the majority of the students explain the concept of global citizenship mostly through "identity and belonging, responsibilities and rights". It can be said that students adopted the concept of global citizenship not as a structure limited to acquiring

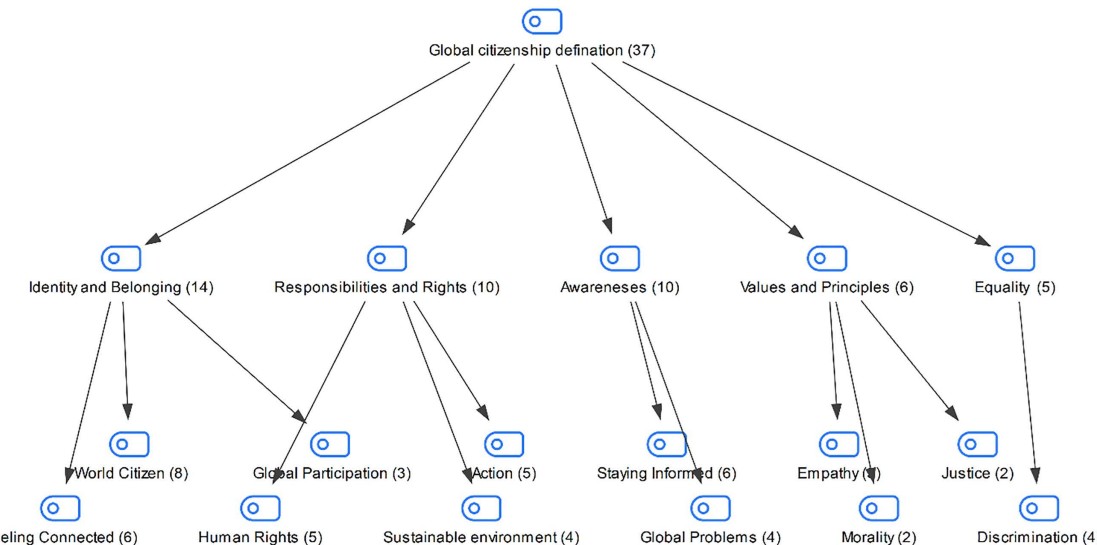

**Fig 1. Maxmap of participants' definitions of global citizenship.**

knowledge or intercultural awareness, but as a more holistic approach such as individuals taking action, producing values, and taking social responsibility.

Some of the students' responses are as follows:

"Someone who understands the world and feels part of it, no matter the nation and race. A person who feels like they belong with everyone else in the world socially, politically, economically etc." (F8)

"Global citizenship is the recognition that we are all part of a shared humanity, with a responsibility to respect diversity, act ethically, and contribute to the global community's well-being." (M12)

"I believe global citizenship refers to people who are aware of common issues about the world, not only their own countries." (F20)

### Findings on whether students see themselves as global citizens

In the interview conducted with the students in the study group, they were asked whether they saw themselves as global citizens. The students' responses are shown in Fig 2:

When the students' answers to the question of whether they see themselves as global citizens are examined, it is seen that the majority of the students define themselves as global citizens. The rate of those who gave a positive opinion (f 22) shows that the majority of the students are open to the idea of global citizenship. Although this shows that the students are inclined to adopt the global citizenship identity, it can be said that some of them lack active participation. The rates of those who gave a partial answer (f 9) and a negative answer (f 6) draw attention to the lack of awareness, knowledge or conflicts between national identity and global identity. The group that gave a partial answer may have given this answer due to a lack of national identity, ethical responsibility or awareness, although they are open to the concept of global citizenship. It can be said that the group that gave a negative opinion cannot establish a connection with global citizenship due to lack of knowledge, national identity preference or ethnocentric views.

Some of the students' responses are as follows:

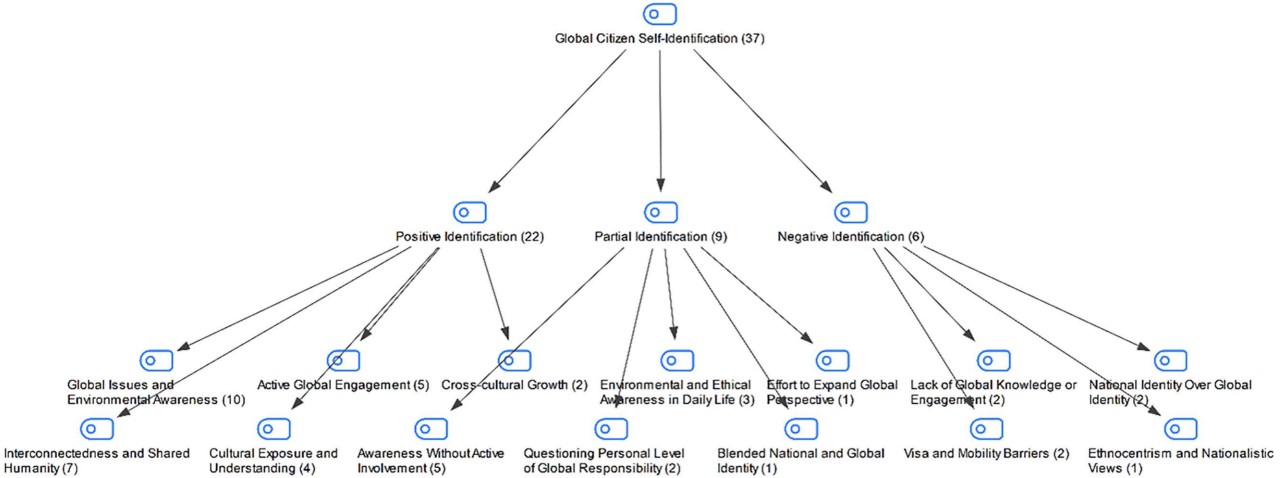

**Fig 2. Maxmap of whether the participants see themselves as global citizens.**

"Yes, because I actively engage in issues that affect the world beyond my immediate environment, seek to understand diverse perspectives, and aim to contribute positively to global challenges." (M9)

"I am aware of world's problems, like pollution, global warming but I haven't done anything about that, didn't participate any activity." (F17)

"No, I don't. Because I have a strong identity of citizen of my country." (F2).

## Findings on the skills, knowledge and attitudes that global citizens should have, according to students

In the interview conducted with the students in the study group, the skills, knowledge and attitudes that global citizens should have been asked. The students' responses are shown in Fig 3:

An examination of students' responses regarding the skills, knowledge, and attitudes required for global citizenship reveals that they primarily emphasize communication, critical thinking, and empathy. This emphasis on these skills may indicate that they prioritize global citizenship from both a practical and social perspective. Students identified global awareness and sustainability as essential knowledge for global citizenship. Students' attention to global issues and sustainability can be interpreted as an awareness of global problems. Furthermore, students' attitudes suggest values such as responsibility, ethical awareness, and respect for diversity. These responses may reflect students' sensitivity to justice and equality. Based on these findings, it is possible to say that students approach the concept of global citizenship in a multidimensional manner and are aware of both individual and societal responsibilities.

Some of the students' responses are as follows:

"Acceptance of different cultures, understanding issues on global level and self-identification of one a global citizen (I think it's a must to understand we are all the same people in the world, just with differences that should not matter)." (M10)

"For me global citizen should have critical thinking, empathy, cultural awareness, adaptability, and a commitment to social justice are essential for global citizenship, as they enable individuals to engage constructively in global issues." (M17)

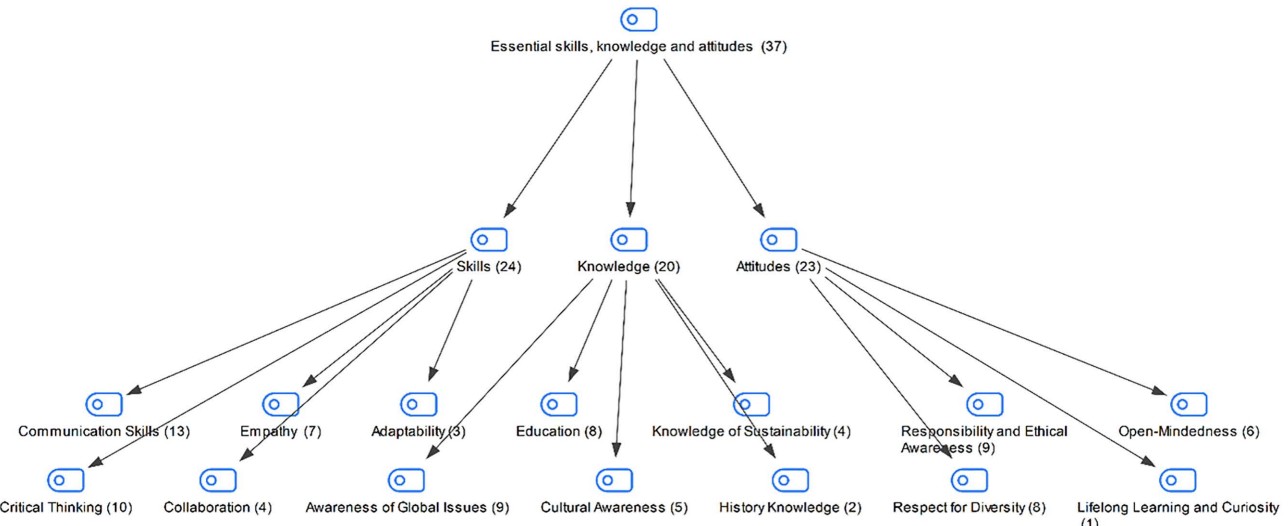

**Fig 3. Maxmap of the skills, knowledge and attitudes that global citizens should have according to the participants.**

"Someone who is tolerant, open-minded and respectful of the social community's way of life should be someone who thinks that people are of the world and belong to the world, and accepts the earth as a whole without borders and cares about sustainability in every action." (F4)

### Findings on students' responses to the responsibilities of global citizens to promote prosperity and justice in the world

In the interview conducted with the students in the study group, they were asked about the responsibilities of global citizens to promote prosperity and justice in the world. The students' responses are shown in Fig 4:

When the students' responses regarding the responsibilities of global citizens to promote prosperity and justice in the world are examined, the students stated that global citizens should generally be advocates of justice, have cultural empathy and be sensitive to the environment in order to promote prosperity and justice in the world. The students addressed the responsibilities of global citizens from various perspectives and emphasized the issues of advocacy of justice, cultural empathy and environmental responsibility the most. It's safe to say that students' statements focused on global citizenship responsibilities, such as protecting justice, fostering intercultural understanding, and demonstrating environmentally conscious attitudes. In this context, students emphasized the importance of individuals assuming social, cultural, and environmental responsibilities to promote prosperity and justice.

Some of the students' responses are as follows:

"Understanding of global issues and participating in fixing those (actively and passively). Empathy for everyone no matter their culture, religion or race." (M11)

"Individuals should strive to promote social justice, protect the environment, and support global cooperation to enhance prosperity and fairness worldwide." (M3)

"Being an active citizen, raising their voice when needed, being educated." (F7)

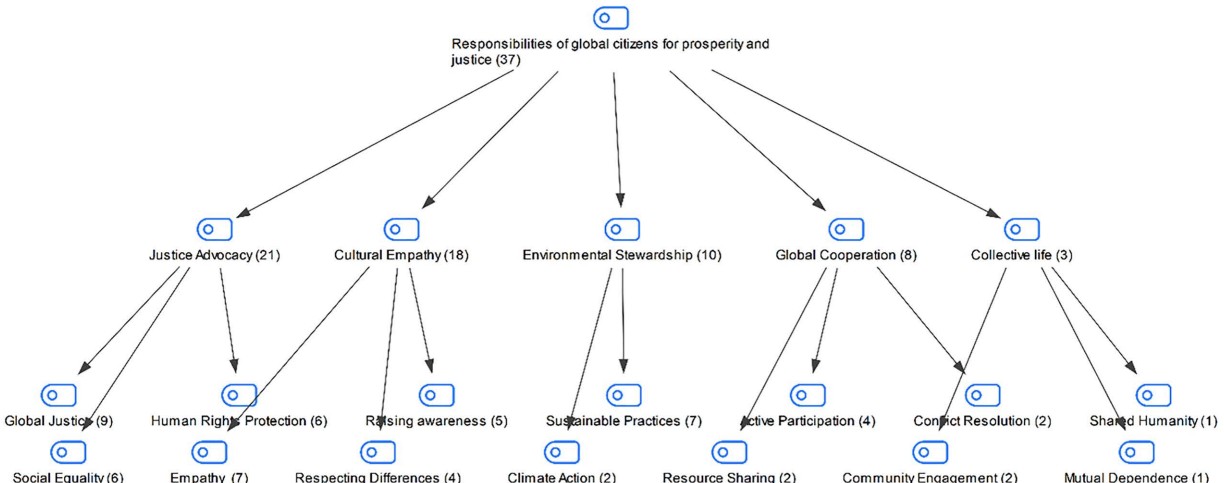

**Fig 4. Maxmap of participants' responses regarding the responsibilities of global citizens to promote prosperity and justice in the world.**

### Findings regarding students' responses to the question "the world's biggest problems"

In the interview conducted with the students in the study group, the biggest problems in the world were asked. The answers of the students are shown in Fig 5:

When the answers given by the students about the biggest problems in the world are examined, the answers given by the students about the biggest problems in the world are gathered under the themes of environmental problems and climate change, social injustice and ethical problems, conflict and violence, health and economic difficulties. According to these themes, it is possible to say that students have awareness of multidimensional problems encountered on a global scale, that they have developed sensitivity to these problems both at individual and societal levels, and that they can evaluate world problems comprehensively from different perspectives.

Some of the students' responses are as follows:

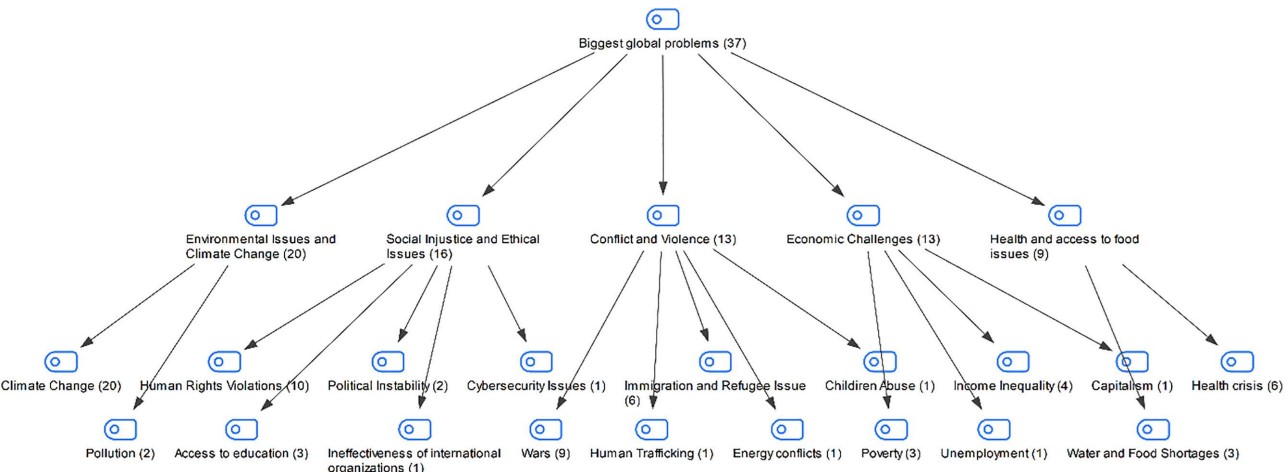

**Fig 5. Maxmap of the participants' responses to the question of the biggest problems in the world.**

"The biggest global problems include climate change, income inequality, and political instability." (F12)

"Climate change, health crisis, inequality, conflict and violence." (F13)

"Injustice in global economy wars because they affect civilians and global warming." (M25)

**Findings regarding students' question on what sustainability means to them**

In the interview with the students in the study group, they were asked what sustainability meant to them. The students' answers are shown in Fig 6:

Students' responses to the question about what sustainability means to them appear to fall under the themes of "efficient and conscious resource use, circular and holistic sustainability, intergenerational environmental responsibility, and collective-global action." The themes of "efficient and conscious resource use" and "circular-holistic sustainability" highlight the material dimension of sustainability, emphasizing the concrete form of environmental responsibility through technical issues such as energy, waste management, and recycling. In contrast, the theme of "intergenerational environmental responsibility" defines sustainability not only in terms of the well-being of present but also of future generations, demonstrating a value-based approach. Sub-concepts such as sustainable education and a livable future under this theme highlight the pedagogical and ethical aspects of sustainability. On the other hand, the lesser emphasis on the theme of "collective and global action" suggests that sustainability is largely limited to what individuals and local communities can achieve. It appears that broader issues such as international cooperation and the shaping of international decisions or policies have not been adequately addressed. When the themes are examined, it is seen that the concept of sustainability focuses more on daily practices, individual responsibilities and practical solutions, while broader issues such as structural change, international cooperation and political decision processes are less emphasized.

Some of the students' responses are as follows:

"For me sustainability is a balanced approach to environmental health, social equity and economic viability." (F14)

"Sustainability means ensuring that we take care of the environment enough to make sure that future generations can live in a safe world." (M1)

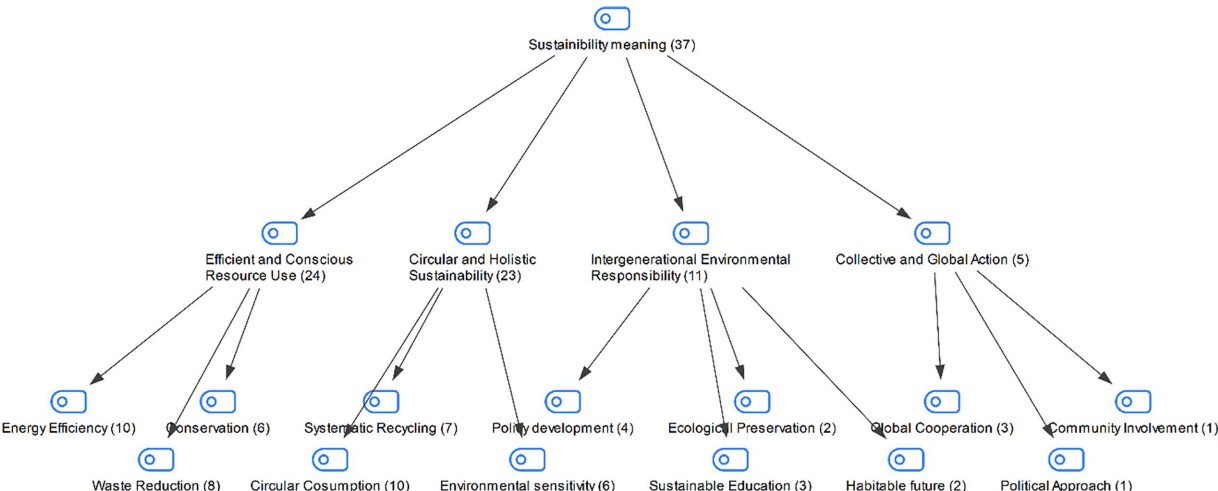

**Fig 6. Maxmap of participants' views on what sustainability means to them.**

"Sustainability means ensuring that we take care of the environment enough to make sure that future generations can live in a safe world." (F12)

## Findings on students' actions in their daily lives for a sustainable world

In the interview conducted with the students in the study group, they were asked about the actions they take in their daily lives for a sustainable world. The students' answers are shown in Fig 7:

When the answers given by the students to the question about the actions they take in their daily lives for a sustainable world are examined, we can say that the students generally carry out actions such as sustainable consumption habits, waste management and recycling, and conservation of resources in their daily lives for a sustainable world. Based on the responses, it can be said that students' individual contributions to sustainability are not limited to consumption or waste management, but also encompass a wide range of areas, including social awareness, environmental advocacy, resource efficiency, and transportation habits. Student responses also demonstrate that sustainability is not limited to specific areas; individuals can contribute to sustainability by acting environmentally consciously in many different areas of daily life.

Some of the students' responses are as follows:

"I do not use water and electricity unnecessarily. I compost food waste in the garden. I throw plastic bottle paper waste into recycling bins. By following world developments on social media and newspapers, I tell and offer people who are around me world should be without war and more clean." (F5)

"Consuming less and more intentionally, using green energy sources and talking public transportation." (F14)

"I minimize waste, reduce energy and water consumption, support sustainable products, and engage in discussions and actions that promote environmental and social responsibility." (M14)

## Findings regarding students' responses to the impact of global citizenship on sustainability

In the interview conducted with the students in the study group, the impact of global citizenship on sustainability was asked. The students' responses are shown in Fig 8:

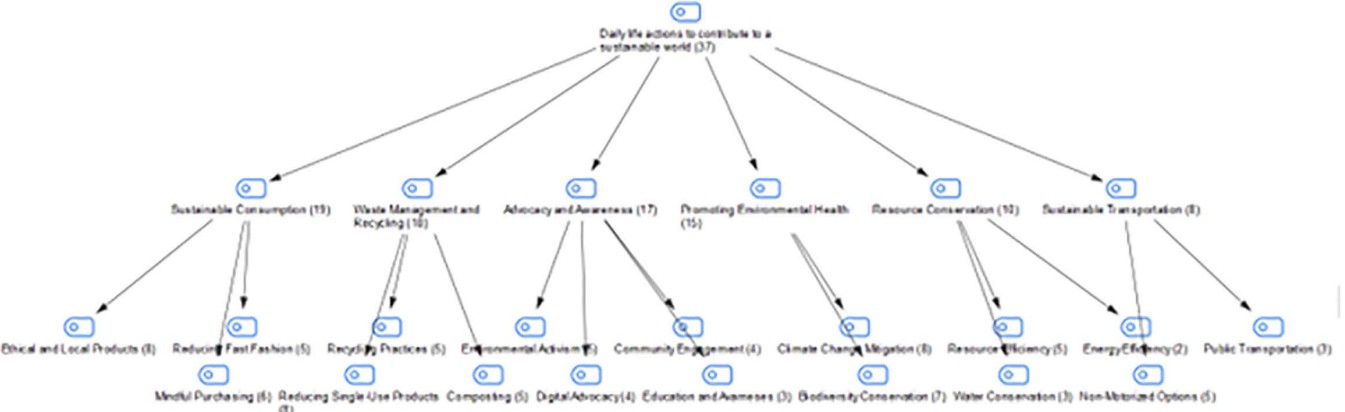

**Fig 7. Maxmap of the actions taken by the participants in their daily lives for a sustainable world.**

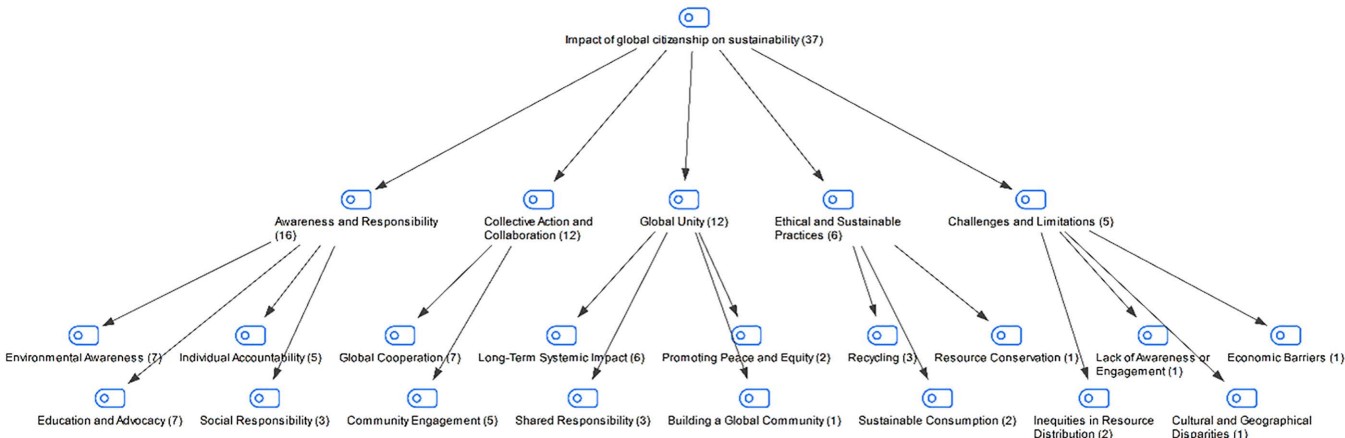

**Fig 8. Maxmap of participants' views on the impact of global citizenship on sustainability.**

When we look at the students' answers to the question about the impact of global citizenship on sustainability, we can say that the students generally explain the impact of global citizenship on sustainability under the themes of awareness and responsibility, global unity, cooperation and joint action, ethical and sustainable practices, challenges and limitations. The students have addressed the impact of global citizenship on sustainability quite comprehensively with their answers. This shows that they have a conscious and responsible global perspective. In addition, while emphasizing positive aspects such as awareness, collective cooperation, and ethical practices, the students also drew attention to challenges such as resource inequality and economic barriers. In short, the students evaluated the impact of global citizenship on sustainability in a balanced way, both in terms of opportunities and limitations. It is possible to say that this shows that they address global problems from a multidimensional perspective.

Some of the students' responses are as follows:

"Global citizenship encourages individuals to see beyond borders and work collaboratively to address issues like climate change and inequality, amplifying efforts towards a more sustainable future." (M14)

"The impact of global citizenship on sustainability can be assessed by looking at how individual and collective actions contribute to sustainable development goals (SDGs) and positive environmental and social outcomes." (F3)

"Global citizenship fosters a sense of shared responsibility, encouraging collective actions toward environmental and social sustainability." (M8)

**Findings on the decisions and actions students would make/take if they were un secretary**

In the interview conducted with the students in the study group, they were asked about the decisions and actions they would make/take if they were UN secretary. The students' answers are shown in Fig 9:

When students' responses to questions about the decisions and actions they would take if they were Secretary-General of the United Nations were examined, it was observed that they included topics such as climate action and environmental policies, education and public awareness, sustainable resource use, social and economic justice, and innovative practices. These responses suggest that the students addressed environmental and social issues together and sought to develop a sensitivity to these areas. The fact that students also prioritized issues such as environment, equality, and education suggest that they tend to address sustainability issues from various perspectives.

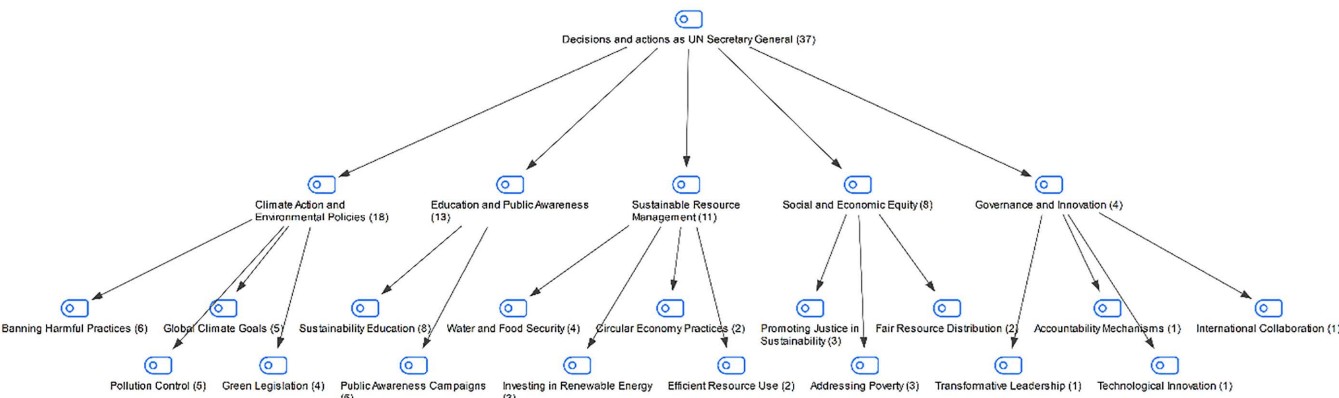

**Fig 9. Maxmap of participants' definitions of global citizenship.**

Some of the students' responses are as follows:

"Advocate for more plastic bans and try to create sustainable plans for regions that are hit with poverty, in a sense that it's not just one-time solution, but a long term one (giving food will not solve the poverty as we need to think how to create conditions for the people there to live a proper life, where they can themselves create work, food and every condition we have here in more developed regions)." (F16)

"As UN Secretary General, I would prioritize global collaboration to enforce sustainable policies, such as carbon reduction goals and equitable resource distribution." (M6)

"I would stop sending trash to other countries. This is a common trade in today's word, and it is a huge threat to sustainability." (F21)

## Conclusion and discussion

The purpose of this study is to examine the global citizenship tendencies of international university students studying in Turkey and to explore their relationship with sustainability. The study group's overall global citizenship tendency was moderate, indicating that participants demonstrated an interest in and a sense of responsibility toward global issues, although this awareness was not particularly strong [50–55]. Liu et al. [56] reported that Taiwanese university students generally exhibited high levels of self-awareness regarding global citizenship.

Participants scored moderately across all dimensions of the Global Citizenship Scale, although significant variability was observed among the different dimensions. Relatively higher (yet still moderate) scores were found in social justice, environmental sustainability, valuing diversity, and intergroup cooperation, indicating sensitivity to justice, environmental protection, diversity, and collaboration. In contrast, scores for ideal environment, global awareness, global citizenship identity, responsibility for action, and intergroup empathy were moderate but comparatively lower. This suggests that the depth of knowledge and influence may be limited, global identity is not fully consolidated, and levels of action lag behind awareness and empathy.

Regional comparisons revealed that the European group had the highest mean score, consistent with the region's strong emphasis on democratic infrastructure, education, environmental policies, human rights, and international cooperation. The Asian and Sub-Saharan African groups exhibited intermediate levels, likely reflecting heterogeneity in

development, inequality, environmental challenges, and civic mobilization dynamics. The Middle East and North Africa, as well as the Russia/Eurasia groups, received lower scores, a pattern that may be linked to political instability, limited civic structures, and weaker international cooperation.

Although direct regional-scale studies are limited, findings at the country level generally align with this pattern. Szelenyi and Rhoads [57] reported higher levels of global identity among Italian students compared to their Brazilian and Chinese peers. Der-Karabetian, Cao, and Alfaro [58] found that national identity was more prominent among Chinese students, whereas global identity was more pronounced among Taiwanese students. Katzarska-Miller et al. [59] observed a strong global citizenship identity in both Bulgaria and India, though the emphasis on values differed between the two groups. In contrast, Gerodimos et al. [54] reported that students of African and Asian origins experienced globalization more intensely and assumed greater responsibilities than those of American and Western European origins.

Social media use was positively and significantly correlated with ideal environment, global awareness, global citizenship identity, intergroup empathy, valuing diversity, social justice, environmental sustainability, intergroup cooperation, responsibility for action, and the overall score. This finding underscores the role of social media in shaping these dispositions. Similar correlations have been reported by Cleofas and Labayo [60], Ekici [61], Ben Ltaifa and Derbali [62], Lee, Baring, Maria, and Reysen [63], Çakmak, Bulut, and Taşkıran [64], and Maguth and Yamaguchi [65].

No significant differences were found between genders for most dimensions—including global awareness, value of diversity, and social justice—except for a small effect favoring men on the dimension of intergroup empathy [50,65–68]. Conversely, some studies report higher scores for women [54,69]. Regarding age, participants aged 27 and older scored higher on the dimensions of intergroup empathy, value of diversity, and social justice; these findings align with those of Sherman [70]. However, other studies report no effect [66,67] or a negative relationship between age and global citizenship [71].

Participation in global events and membership in international associations were associated with higher scores across all dimensions, underscoring the importance of experiential learning and civic engagement [67,72–74]. The validity and reliability of the scale used have been demonstrated in various contexts [75–77], enhancing the comparability of the findings. This study focuses on the trends among international students in Turkey and their implications for the pursuit of sustainable global solutions.

Qualitative findings indicate that students conceptualize global citizenship in a multidimensional manner, encompassing aspects of identity, belonging, awareness, responsibility, equality, and justice. Previous studies have identified similar themes: Aydın and Andrews [78]; Aydın et al. [79]; Altıkulaç and Yontar [80]; Kuleta-Hulboj [81]; Massey [82]. The proportion of those who identify as global citizens is high. This finding is consistent with WVS [83], Liu et al. [56], and Baker and Fang [84]. Participants who do not identify as global citizens or are undecided often attribute this to a lack of knowledge/awareness and tensions between national and global identities [85,86].

In terms of skills, knowledge, and attitudes, students emphasized communication, empathy, critical thinking, awareness of global issues and sustainability, as well as ethical sensitivity and respect for diversity. This emphasis is consistent with Oxfam's framework [87], Hunter's Global Competency Model [88], and findings from preservice teachers [29]. Furthermore, in the Responsibilities (well-being and justice) area, themes centered around justice advocacy, cultural empathy, and environmental awareness, consistent with the classifications put forward by Oxfam [89] and UNICEF [90].

Students identified environmental and climate issues, social injustice, conflict and violence, as well as health and economic hardships as major global challenges. This finding reflects a high level of awareness and a solution-oriented orientation [91–94]. Sustainability awareness extends from individual behaviors to policy-level considerations, often emphasizing resource efficiency, recycling, waste management, and sustainable consumption. These results align with the findings of Behm [95], Owens and Legere [96], and Demirci and Teksöz [97]. The reported daily practices—such as recycling, waste management, and resource conservation—are similarly observed among preschool teachers [98], high school students [99], and larger samples [100].

Students discussed facilitators such as ethical practice, cooperation, and awareness in the relationship between global citizenship and sustainability, as well as constraints including resource inequality and economic hardship. Furthermore, a positive correlation was identified between increased global citizenship and higher levels of sustainable development [69].When asked about priorities at the United Nations level, climate action, education, social equity, and innovation were emphasized; these priorities reflect the need to balance environmental and social dimensions in accordance with Our Common Agenda [101].

As a result, international students generally demonstrate moderate levels of global citizenship, characterized by a multidimensional understanding, broad awareness, values-aligned attitudes, and daily sustainability practices. These tendencies support solution-focused approaches and contribute to shaping the students' future potential as global citizens.

## Results

When the quantitative and qualitative findings of the study are evaluated together, it becomes evident that international students possess a global citizenship awareness. The quantitative data indicate a moderate level of this awareness, whereas the qualitative findings reveal a higher and more multifaceted understanding. Both data types highlight a growing consciousness of environmental and social issues. While the quantitative results emphasize the influence of social media and increased participation in global activities and organizations, the qualitative findings associate this awareness with empathy, communication, and ethical sensitivity, as well as a process rooted in solidarity and collaboration. There is a clear convergence indicating that active participation enhances awareness. The simultaneous integration of these findings strengthens the study's scope and reliability.

While global citizenship awareness is generally shaped by geographical, cultural, and social contexts, the understanding of universal responsibility is strengthened. This concept is particularly evolving, encouraging students to take a more active role in social and environmental issues. Consistent with the sub-dimensions of the scale used, the study focused on moral, social, and environmental aspects. Values such as human rights, social justice, and environmental sustainability were evident in students' emphasis on empathy, responsibility, and sensitivity to nature.

As Torres [102] notes, sustainable development is the "twin sister" of global citizenship education; both must be addressed together to achieve the seventeen Sustainable Development Goals. The 0.56% decline in peace levels in 2024, as reported by the Global Peace Index, and the theme of the Brazilian G20 Presidency [103], "Building a Just World and a Sustainable Planet", this necessity. Expanding global citizenship is essential for fostering cooperation and creating a more peaceful world.

The study offered an integrated perspective, with scale means representing general attitudes and interviews providing experience-based explanations of these attitudes. Social media engagement, event participation, and association memberships are all linked to these attitudes. Older age groups and students of European origin stand out somewhat due to their greater opportunities for cultural interaction. Participants' understanding of responsibility, centered on justice, equality, and environmental awareness, demonstrates that global citizenship requires action beyond mere conceptualization. Therefore, it is important to design plans and activities for international students that encourage both knowledge transfer and behavioral engagement.

The unique contribution of this study lies in its systematic comparison of international students studying in Turkey across different geographic origin groups (Europe, Asia, Sub-Saharan Africa, the Middle East and North Africa, and Russia/Eurasia) within a single sample. This design enables comparisons across regions within the same country, eliminating the influence of country-specific factors and allowing for clearer identification of regional differences. The findings offer a direct foundation for developing region-sensitive, dimension-based, and evidence-based interventions in university policies and course design.

## Limitations

The most significant limitation of this study is the small sample size of the qualitative interviews. A more comprehensive exploration of the research findings through qualitative interviews would have been beneficial, especially given that global citizenship tendencies vary across regions such as Europe, Asia, and Sub-Saharan Africa. Additionally, the study did not include international students from the United States, Australia, or developed Asian countries due to the low number of students from these regions at the participating universities and their reluctance to accept work offers. Furthermore, although the quantitative findings indicated that age and social media influence global citizenship tendencies, with gender having a minimal effect, these results could not be corroborated through the qualitative interviews. This limitation arose because only students who were available to participate in the interviews were included, making it difficult to examine these variables qualitatively. Moreover, while the qualitative data suggested that students experienced conflicts between global and national identities, this important issue could not be explored in depth.

## Recommendations

In line with the research findings, future studies should primarily expand the sample size by including students from diverse geographic regions to address the limitations identified. Accessing students from previously inaccessible areas will enrich the discussion and enhance the generalizability of the results. To compare the variables identified in the quantitative findings, confirmatory qualitative interviews should be conducted to enable a comparative analysis of quantitative and qualitative data. Additionally, experimental studies examining the effectiveness of global citizenship and sustainability-focused content in university curricula, as well as follow-up studies covering the post-graduation period, could be undertaken to assess the long-term impact of these interventions. Comparative studies across different country contexts would also be valuable in elucidating the relationship between the findings and contextual variables.

Future studies should also examine in depth the potential conflicts students experience between their global and national identities. The research findings indicate that while global citizenship awareness appears moderate in quantitative data, it is higher in qualitative data; the reasons for this discrepancy—such as lack of knowledge or identity conflict—should be explored thematically. Additionally, the ways in which global citizenship tendencies are shaped by cultural and political contexts should be examined through comparative analyses across different countries and institutions. The impact of digital and network-based factors—such as social media, participation in global events, and membership in global organizations—on identity formation and the strengthening of these tendencies should also be explored in depth as a distinct theme.

Various studies should aim to address knowledge gaps and manage the conflict between global and national identities by designing educational interventions that help students unlock their potential. Integrating global citizenship and sustainability-focused content into university curricula can concretely connect these concepts to education. The conscious and purposeful use of social media and other digital tools, opportunities for intercultural interaction, and the creation of experiential learning environments can help students translate their awareness into action in both daily life and global challenges. Finally, given their young and dynamic nature, planned support for this group is recommended to reduce inequalities and raise global awareness.

The results of this research highlight the necessity of strategic interventions to enhance students' awareness of global citizenship within education policies and global cooperation initiatives. Future research in this area will contribute to advancing the understanding of global citizenship for a sustainable world.

## Author contributions

**Data curation:** Adem Uzun, Osman Akhan.

**Formal analysis:** Osman Akhan.

**Investigation:** Adem Uzun, Osman Akhan, Ali Özkaya.

**Methodology:** Osman Akhan, Ali Özkaya.

**Resources:** Osman Akhan, Ali Özkaya.

**Supervision:** Adem Uzun.

**Validation:** Saim Turan.

**Visualization:** Saim Turan.

**Writing – original draft:** Osman Akhan, Saim Turan, Ali Özkaya.

**Writing – review & editing:** Osman Akhan, Saim Turan, Ali Özkaya.

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
