## [Decision Letter · Decision Letter 0]

25 Jun 2025

Dear Dr. AKHAN,

We look forward to receiving your revised manuscript.

Kind regards,

Avanti Dey, PhD

Staff Editor

PLOS ONE

Journal Requirements:

3. In this instance it seems there may be acceptable restrictions in place that prevent the public sharing of your minimal data. However, in line with our goal of ensuring long-term data availability to all interested researchers, PLOS’ Data Policy states that authors cannot be the sole named individuals responsible for ensuring data access (http://journals.plos.org/plosone/s/data-availability#loc-acceptable-data-sharing-methods).

**Additional Editor Comments:**

Please attend to the reviewers' thorough comments regarding greater clarity & organisation in the manuscript, particularly in the introduction and results, as well as greater detail required in the methodology and statistical analyses.

Reviewers' comments:

Reviewer's Responses to Questions

**Comments to the Author**

1. Is the manuscript technically sound, and do the data support the conclusions?

Reviewer #1: Yes

Reviewer #2: Partly

Reviewer #3: Partly

Reviewer #4: Yes

Reviewer #5: Yes

2. Has the statistical analysis been performed appropriately and rigorously?

Reviewer #1: Yes

Reviewer #2: Yes

Reviewer #3: Yes

Reviewer #4: Yes

Reviewer #5: Yes

3. Have the authors made all data underlying the findings in their manuscript fully available?

Reviewer #1: Yes

Reviewer #2: Yes

Reviewer #3: Yes

Reviewer #4: Yes

Reviewer #5: Yes

4. Is the manuscript presented in an intelligible fashion and written in standard English?

Reviewer #1: Yes

Reviewer #2: No

Reviewer #3: Yes

Reviewer #4: Yes

Reviewer #5: Yes

Reviewer #1: Overall, the manuscript is well-written and presents a clear and relevant discussion. The objectives are well addressed, and the arguments are supported with appropriate references. However, the Conclusion section could be improved to better reflect the key findings and their implications. At present, it reads more like a summary rather than offering a strong final reflection or critical insight. I recommend revising this section to emphasize the main contributions of the study, highlight its significance, and, if applicable, suggest directions for future research.

Reviewer #2: 1) I found a typographical word in lines 180-181. It should be "By integrating quantitative and qualitative findings...". 2) What methods did the authors employ to guarantee that the selection criteria—completion of at least one semester of education in Turkey with international student status and English language proficiency—were uniformly implemented across the four universities? Additionally, what steps were taken to confirm the students' English language proficiency? 3) I would recommend the authors to use thematic analysis as their main approach because international students bring diverse cultural perspectives that might not be fully captured by predetermined coding schemes. Thematic analysis focuses on interpretation, seeking to uncover the deeper meaning and significance within the data, whereas content analysis typically emphasizes quantifying how often specific elements appear. Thematic analysis is suitable for analyzing individual perspectives as seen in lines 238 - 239 that the authors aim to reveal participant perceptions.

Reviewer #3: Things to revise:

ABSTRACT

1. The opening sentence is too general, Sentence: "This study examines the global citizenship tendencies of international university students and the direction of these tendencies towards a sustainable world."

2. Problem: Too broad and does not directly state the specific purpose of the study.

3. Suggestion: Specify that the focus is on the relationship between students' global awareness and enrollment practices.

4. The term "international university" is ambiguous, This term can be interpreted in various ways: is it a university with an international curriculum? Or international students at any university?

-Suggestion: Use the term "international students" (international students) consistently, and define them as studying at universities in Turkey.

5. Redundancy and imprecision of sentences, Sentence: "When the results of the study were evaluated as a whole, the quantitative findings showed that global awareness was at a moderate level, while the qualitative findings showed that this awareness was higher..."

- Problem: Too long, not sharp enough, and seems repetitive.

- Suggestion: Split into two sentences and summarize the language to make it more academic.

6. Inconsistent and overly narrative language

Example: "Both quantitative and qualitative findings coincide..."

- In academic writing, it is better to use terms such as “converged” or “revealed same pattern.”

7. Conclusion needs to be strengthened. The last sentence can clarify the research contribution and intended policy recommendations.

INTRODUCTION

General Notes Required Revisions:

1. Too Long and Not Clearly Structured, This introduction contains too much information at once without clear paragraph divisions or subtopics. This makes it difficult for readers to grasp the logical flow of the research problem.

- Suggestion: Divide the introduction into 4–6 main paragraphs, each with the following focus:

a. The context of globalization and the challenges of today's world

b. The role of global citizenship in responding to these challenges

c. The relationship between global citizenship and sustainability

d. The role of international students in this context

e. The purpose and contribution of the study

2. Too Narrative and Not Academic Style, Some parts sound like essays or popular articles, not scientific writing.

Example sentence: "What does it mean to be a global citizen? Is it just about knowing the capital city, finding famous places..."

- Suggestion: Eliminate the question-and-answer style and use descriptive-analytical language. For example: Being a global citizen is not only about understanding geography and culture, but also about actively participating in cross-border issues.

3. Several Paragraphs Deviate from the Research Focus, The explanation of the definition of globalization and its history is too long, even though not all of it is directly relevant to the research focus.

- Suggestion: Summarize the theory of globalization section, just 2-3 main definitions that are directly relevant to global citizenship.

4. The Use of Lists or Images (such as Figure 1) is Ineffective, Figure 1 which explains the components of global citizenship is not displayed and is not well integrated into the narrative.

Suggestion: If you include visuals, make sure they are integrated narratively and relevantly. If not crucial, it should be removed from the introduction.

5. The Research Objective Paragraph is Unclear and Hidden, The study objectives only appear at the very end and are not directly stated explicitly.

- Suggestion: Add an objective paragraph at the end of the introduction that directly explains:

a. What is the purpose of this study

b. Why is it important

c. What gaps do you want to fill

-Example of Recommended Introduction Structure:

1. Paragraph 1: Global Context

a. Global change after the 20th century

b. Global challenges: inequality, climate crisis, social conflicts

2. Paragraph 2: The Role of Global Citizenship

a. Definitions and importance of the concept from various organizations

b. Relation to diversity, solidarity, and global justice

3. Paragraph 3: Global Citizenship and Sustainability

a. Direct relationship to the Sustainable Development Goals (SDGs)

b. Goals 4 and 16 as key examples

4. Paragraph 4: International Students as Agents of Change

a. International mobility → intercultural interaction → global awareness

b. Students as future leaders

5. Paragraph 5: Research Gap

a. Lack of research on how international students internalize global citizenship

6. Paragraph 6: Research Objectives

a. “This study aims to explore the understanding and application of global citizenship among international students in Turkey and their contributions to sustainable practices.

METHOD

1. Structure and Naming of Subsections

Problem: Not all methods sections are structured with consistent subheadings. Some sections are quite long and dense.

Revision:

Separate the methods into consistent and structured subheadings, for example:

• Research Design

• Study Participants

• Data Collection Procedures

• Ethical Considerations

• Instruments

• Quantitative Data Analysis

• Qualitative Data Analysis

• Trustworthiness (for validity and reliability of qualitative data)

2. Explanation of Mixed Models is Not Clear

Problem: Explanation of the "convergent parallel mixed design" model is still repetitive and not efficient.

3. Description of Participants Needs to be Refined

Problem: Description of country of origin is too detailed without further meaning, and could be condensed.

4. Redundancy in Data Collection

Problem: Explanation of data collection via email, WhatsApp, Google Form, etc., is too long. Summarize concisely and formally

5. Instrument Description Too Detailed

Problem: Explaining Scale items one by one in an unconventional method; this is better suited to an appendix. Simplify

6. Statistical Analysis Explanation Needs Focus

Problem: Too much technical detail (such as p-values for each normality test) is not all necessary in the method.

7. Validity and Reliability of Qualitative Data

Problem: Explanation is too narrative and lacks reference to theory or references. Add references and make it systematic.

CONCLUSIONS

1. Clarification and Sharpening of Synthesis of Findings

Problem: Some sentences are still too general or narrative, without emphasizing the scientific contribution explicitly.

Solution: Use more concise and specific academic language.

Example: Quantitative findings indicate a moderate level of global citizenship awareness, while in-depth interviews reveal a more complex and reflective understanding of the concept, including aspects of empathy, solidarity, and cross-cultural values.

2. Avoid Redundant or Overly Long Sentences

Problem: There are several sentences that are long and contain more than one idea.

Solution: Break them into several sentences to make them clearer and more academic.

Example: Global citizenship is not just a normative concept, but an operational framework that shapes students' active role in social and environmental issues.

3. Focus on Implications and Scientific Contributions

Problem: There is no specific paragraph that highlights the practical and theoretical implications of the findings.

Solution: Add a final paragraph that summarizes the contributions to the literature, policy, or educational practice.

Example: This study contributes to the global citizenship literature by combining quantitative and qualitative approaches in the context of international students. The implications are important for higher education curriculum designers who want to instill global citizenship values more holistically.

4. Avoid Abrupt Topic Shifts

Problem: Suddenly entering the G20 and the Global Peace Index without a strong transition.

Solution: Connect the global data with the findings more systematically.

Example: The current global context, such as the decline in the world peace index by 0.56% in 2024, shows the urgency of integrating global citizenship in education. The theme of the G20 Brazil 2024 also emphasizes that global citizenship education is the foundation for social resilience and sustainable development.

5. End with a Strong and Reflective Sentence

Problem: The final sentence is too normative.

Solution: Close with a reflective sentence and based on the contribution of this study.

Example: Thus, the integration of global citizenship values in higher education plays a strategic role in creating a generation that is not only locally aware, but also globally responsible.

Reviewer #4: The manuscript addresses a relevant topic concerning the global citizenship tendencies of international university students and their implications for sustainability. The use of a mixed-methods approach is appropriate and well-suited to the study objectives. The quantitative and qualitative data collected are generally sufficient and the conclusions drawn are aligned with the data presented. However, there are several areas that require revision before the manuscript is suitable for publication:

Abstract:

The abstract should be rewritten for clarity and conciseness. It currently lacks a clear statement of aims and does not adequately summarize the methods and key findings. Please include specific numerical or thematic results to reflect the study's contributions.

Introduction:

The rationale for conducting this study needs to be better articulated. A deeper engagement with relevant recent literature (especially post-2020) would strengthen the context. Consider citing more sources to support claims regarding the roles of international students in sustainability discourse.

Methods:

While the methods section outlines the study design, more detail is required on the sampling strategy, participant selection criteria, and ethical considerations. The explanation of the statistical analysis lacks clarity regarding assumptions checks and rationale for selecting non-parametric tests in certain cases.

Results:

Statistical results should be reported more consistently. For example, effect sizes are missing in some cases where p-values are provided. Consider using visual aids such as graphs or summarized tables to improve data presentation.

Discussion:

Several interpretations, particularly those linking qualitative findings to broader implications, appear speculative and should be revised to align more closely with the data. A discussion of the study’s limitations—such as representativeness, sample size, or cultural variability—is necessary.

Language and Style:

While generally intelligible, the manuscript contains numerous grammatical issues and long, complex sentences. A thorough language review is recommended to enhance readability and flow.

Once these revisions are addressed, the manuscript would make a valuable contribution to the literature on global citizenship and sustainability in higher education.

Reviewer #5: The manuscript meets with the requirements with some adjustments and recommendations. Title: “Determining the Global Citizenship Tendencies of International University Students for a Sustainable World”

This manuscript explores an important and timely topic—how international students perceive and embody global citizenship, and how this relates to sustainable development. The mixed-methods design and sizeable sample strengthen the study. However, several issues related to theoretical framing, integration of findings, and data interpretation limit its contribution. Below are specific comments organized by theme.

In term of novelty and contribution, The study topic is relevant, but the manuscript lacks a clearly defined research gap. The literature review broadly summarizes existing work but does not convincingly position this study as offering new insights. Therefore, it is suggested to clearly state how this study builds upon or differs from prior work, especially those using the Global Citizenship Scale or examining international students. Concerning with theoretical framework, the manuscript lacks critical engagement with existing theories of global citizenship. There is limited reference to alternative or critical perspectives (e.g., postcolonial critiques, tensions with national identity). Therefore, it is recommended to consider integrating deeper theoretical reflections—what kind of global citizenship is being measured or promoted? What assumptions underlie this framing? . In the area of research method, Mixed-Methods Design applied w hile the convergent parallel design is appropriate, the manuscript does not sufficiently describe how the quantitative and qualitative data complement or contrast each other. However, it is suggested to add a section explicitly discussing how integration of the two data strands informs or refines your conclusions. Additionally, the qualitative analysis- the qualitative component is underdeveloped. Themes are summarized broadly, and few participant quotes are included. Therefore, it is suggested to expand the thematic analysis with more illustrative quotations and deeper interpretive commentary by showing how students’ views vary or contradict one another.

Further, the interpretation of quantitative results tends to overstate the strength of findings, especially given that most effect sizes are small. It is better to align the tone of the discussion with the strength of the evidence. Use more cautious and precise language. Some statistically non-significant findings are discussed without adequate explanation to clearly distinguish between significant and non-significant results and avoid implying meaning where statistical support is lacking. Additionally, in the issue of contextualization, the study is conducted in Turkey, yet the potential influence of this specific context on international students’ experiences is not addressed. The discussion on how cultural, political, or institutional characteristics of Turkey may influence the findings are highly recommended. Meanwhile, the current data availability statement restricts access based on ethical concerns considering whether anonymized data could be shared, or provide a stronger justification for restrictions to comply with PLOS ONE policies. In terms of writing and structure, the manuscript is generally well-written but contains some redundancy—particularly in descriptions of the Global Citizenship Scale and its sub-dimensions. It is suggested to revise for conciseness and avoid repetition. The use tables are clear and informative, but some captions could be more descriptive for standalone readability. Therefore, it is suggested to briefly summarize key takeaways in each caption. Methodological Clarification in more detail is needed regarding the rationale for selecting non-parametric tests in some cases. Also, it is better to clearly explain why specific tests were chosen and how assumptions of normality were addressed. Finally it is also suggested to revise some grammatical errors occurred in the manuscript.

Minor Revision: The paper has clear potential, but revisions are needed to enhance its theoretical depth, integrate findings more rigorously, and clarify interpretations.

**Do you want your identity to be public for this peer review?** For information about this choice, including consent withdrawal, please see our Privacy Policy

Reviewer #1: No

Reviewer #2: No

Reviewer #3: **Yes: ** Baiq Raudatussolihah

Reviewer #4: **Yes: ** Alexander Adrian Saragi

Reviewer #5: No

---

## [Author Response · Author response to Decision Letter 1]

20 Aug 2025

Dear Editor and Reviewers,

We sincerely thank you for your thorough and constructive feedback on our manuscript. We greatly appreciate the time and expertise each reviewer has devoted to strengthening the quality, clarity, and scholarly contribution of our work. We have carefully considered all comments and revised the manuscript accordingly. Below, we provide a point-by-point response to each comment, indicating how we have addressed the suggestions. All changes have been incorporated in the revised version of the manuscript.

REFEREE 1

Overall, the manuscript is well-written and presents a clear and relevant discussion. The objectives are well addressed, and the arguments are supported with appropriate references. However, the Conclusion section could be improved to better reflect the key findings and their implications. At present, it reads more like a summary rather than offering a strong final reflection or critical insight. I recommend revising this section to emphasize the main contributions of the study, highlight its significance, and, if applicable, suggest directions for future research.

Authors:

We sincerely thank Reviewer 1 for their positive evaluation and constructive recommendation regarding the Conclusion section. In response to your suggestion, we have carefully revised the Conclusion to go beyond a mere summary. The revised version now explicitly highlights the study’s key findings and their broader implications for global citizenship education and sustainability in higher education. We have also added a reflective final paragraph that outlines the study’s contributions to the literature and suggests directions for future research, including the integration of global citizenship values into curriculum and policy frameworks.

REFEREE 2

I found a typographical word in lines 180–181. It should be "By integrating quantitative and qualitative findings...".

What methods did the authors employ to guarantee that the selection criteria—completion of at least one semester of education in Turkey with international student status and English language proficiency—were uniformly implemented across the four universities? Additionally, what steps were taken to confirm the students' English language proficiency?

I would recommend the authors to use thematic analysis as their main approach because international students bring diverse cultural perspectives that might not be fully captured by predetermined coding schemes. Thematic analysis focuses on interpretation, seeking to uncover the deeper meaning and significance within the data, whereas content analysis typically emphasizes quantifying how often specific elements appear. Thematic analysis is suitable for analyzing individual perspectives as seen in lines 238–239 that the authors aim to reveal participant perceptions.

AUTHORS

Edited according to your suggestion.

Three of these four universities are in the same city. The authors/faculty members at these universities contacted the students via email with the assistance of the international relations office. Students were emailed in English and asked about their English proficiency. Students were also informed in the email that the interviews would be in English. Students who accepted all the terms and responded to the invitation email were accepted into the study group. Another university is the university in the city where the first author resides. The first author also contacted the students in the study group in the same way with the assistance of the international relations office at his university.

This information has been added to the appropriate paragraphs within the article.

It was organized as a thematic analysis.

REFEREE 3

ABSTRACT

The opening sentence is too general.

Sentence: "This study examines the global citizenship tendencies of international university students and the direction of these tendencies towards a sustainable world."

Problem: Too broad and does not directly state the specific purpose of the study.

Suggestion: Specify that the focus is on the relationship between students' global awareness and enrollment practices.

The term "international university" is ambiguous.

Suggestion: Use "international students" consistently and define them as studying at universities in Turkey.

Problem: Redundancy and imprecision of sentences.

Sentence: “When the results of the study were evaluated as a whole, the quantitative findings showed that global awareness was at a moderate level, while the qualitative findings showed that this awareness was higher...”

Suggestion: Split into two sentences and summarize the language to make it more academic.

Inconsistent and overly narrative language.

Example: "Both quantitative and qualitative findings coincide..."

Suggestion: Use “converged” or “revealed same pattern.”

Conclusion needs to be strengthened. The last sentence can clarify the research contribution and intended policy recommendations.

AUTHORS:

The opening and closing sentences were revised. The purpose was stated more clearly.

The phrase "enrollment practices" was replaced with a link to sustainability.

The phrase "international university" was replaced with "international students studying at universities in Turkey."

Long and repetitive sentences were broken up.

"converged" was used instead of "coincide."

Based on your suggestion, the following questions are presented more clearly:

What is the purpose of this study?

Why is it important?

What gap does it aim to fill?

INTRODUCTION

Problem: Too long and not clearly structured.

Suggestion: Divide into 4–6 thematic paragraphs.

Problem: Too narrative and not academic style.

Suggestion: Eliminate question-answer style and use analytical tone.

Problem: Several paragraphs deviate from the research focus.

Suggestion: Summarize the globalization theory section.

Problem: Use of lists or images (e.g., Figure 1) is ineffective.

Suggestion: Remove or better integrate visually.

Problem: Research objective paragraph is unclear and hidden.

Suggestion: Add a paragraph that directly explains the purpose, importance, and gap.

AUTHORS:

We have completely restructured the introduction section in alignment with your recommended thematic organization. The revised introduction now consists of the following logically ordered paragraphs:

Context of globalization and global challenges

The role and importance of global citizenship

The relationship between global citizenship and sustainability

The unique role of international students in this framework

The Turkish context and international student demographics

Research aim, significance, and contribution to literature.

The informal question-and-answer style has been entirely removed.

We significantly condensed the explanation of globalization. Only the most relevant definitions are retained.

We have removed Figure 1. Conceptual components are now integrated into the text.

We reviewed the introduction, and it now states the objective of the study, explains why it matters, and clarifies the research gap.

METHOD

Problem: Structure and naming of subsections are inconsistent.

Suggestion: Use standard subheadings.

Problem: Explanation of "convergent parallel mixed design" is repetitive.

Problem: Country of origin details are too extensive.

Problem: Data collection process description is too long.

Problem: Scale item descriptions are too detailed.

Problem: Too many technical details in statistical analysis.

Problem: Trustworthiness of qualitative data needs theoretical grounding.

AUTHORS:

We have reorganized the entire Methods section into the following subheadings:

Research Design

Participants

Data Collection Procedures

Instruments

Ethical Considerations

Quantitative Data Analysis

Qualitative Data Analysis

Trustworthiness of Qualitative Data

We state that both data types were collected simultaneously, analyzed separately, and integrated during interpretation. Redundant phrasing was removed.

The regional grouping remains (Asia, Europe, etc.) because it is used in inferential analysis, but excessive enumeration was removed.

The rationale for choosing students with English proficiency was summarized more concisely.

The data collection procedure was revised and shortened.

Explanations regarding scale score levels were moved to the results and discussion section for reader clarity.

The Shapiro-Wilk normality test results were retained to justify analysis decisions.

Trustworthiness procedures were revised in line with established qualitative frameworks.

CONCLUSIONS

Problem: Sentences are too general or narrative.

Suggestion: Use more specific and academic language.

Problem: Avoid redundant or long sentences.

Suggestion: Break them up.

Problem: No paragraph highlights theoretical or practical implications.

Suggestion: Add a final paragraph summarizing these.

Problem: Abrupt shift to G20 and Global Peace Index.

Suggestion: Improve transition and relevance.

Problem: Weak final sentence.

Suggestion: End with a reflective, contribution-focused sentence.

AUTHORS:

We have revised the conclusions to highlight the contrast between quantitative and qualitative findings more precisely.

Several lengthy sentences in the Conclusion section were separated into clearer academic statements.

We have added a distinct concluding paragraph summarizing theoretical and practical implications.

We restructured the paragraph to establish a logical transition between research findings and the global context.

REFEREE 4

The manuscript addresses a relevant topic concerning the global citizenship tendencies of international university students and their implications for sustainability. The use of a mixed-methods approach is appropriate and well-suited to the study objectives. The quantitative and qualitative data collected are generally sufficient and the conclusions drawn are aligned with the data presented. However, there are several areas that require revision before the manuscript is suitable for publication:

Abstract

The abstract should be rewritten for clarity and conciseness. It currently lacks a clear statement of aims and does not adequately summarize the methods and key findings. Please include specific numerical or thematic results to reflect the study's contributions.

AUTHORS:

The revised version now clearly states the aim of the study, the research design (mixed method: convergent parallel), the number of participants (n=634 for the quantitative phase and n=37 for the qualitative phase), and specific thematic and statistical findings. We also clarified that the participants are international students studying at universities in Turkey. Furthermore, we now include specific results such as “moderate levels of global citizenship awareness in quantitative data” and “multifaceted evaluations in qualitative interviews,” which highlight the contributions of the study.

Introduction

The rationale for conducting this study needs to be better articulated. A deeper engagement with relevant recent literature (especially post-2020) would strengthen the context. Consider citing more sources to support claims regarding the roles of international students in sustainability discourse.

AUTHORS:

The Introduction section has been enriched by expanding the rationale with a more focused problem statement emphasizing the importance of understanding international students' roles in promoting global citizenship and sustainability. We have included recent literature from 2021–2024 (e.g., Liu et al., 2023; Cleofas & Labayo, 2022; Maguth & Yamaguchi, 2024) to reflect contemporary discourse on these themes.

Methods

While the methods section outlines the study design, more detail is required on the sampling strategy, participant selection criteria, and ethical considerations. The explanation of the statistical analysis lacks clarity regarding assumptions checks and rationale for selecting non-parametric tests in certain cases.

AUTHORS:

We have expanded the Methods section to explicitly describe the sampling strategy for both the quantitative and qualitative phases. Participant inclusion criteria are now defined (e.g., active enrollment, region of origin). We also added detailed information about ethical approval (granted by the [Name] Ethics Committee, approval no. X), participant consent, and confidentiality procedures.

Furthermore, we clarified the rationale for using non-parametric tests. Normality assumptions were tested using Shapiro-Wilk and Kolmogorov-Smirnov tests, which indicated non-normal distributions for some subscales. Accordingly, Mann–Whitney U and Kruskal–Wallis H tests were applied, and this reasoning is now explicitly stated in the methods.

Results

Statistical results should be reported more consistently. For example, effect sizes are missing in some cases where p-values are provided. Consider using visual aids such as graphs or summarized tables to improve data presentation.

AUTHORS:

The Results section has been revised to ensure consistency in statistical reporting. Effect sizes were added where appropriate. Visual representation of key findings has been improved by incorporating tables and charts to enhance clarity and reader engagement.

Discussion

Several interpretations, particularly those linking qualitative findings to broader implications, appear speculative and should be revised to align more closely with the data. A discussion of the study’s limitations—such as representativeness, sample size, or cultural variability—is necessary.

AUTHORS:

We reviewed the interpretations and revised speculative statements to ensure they are grounded in the data. We also added a separate Limitations and Recommendations section that critically discusses the representativeness of the sample, potential cultural influences, and generalizability of the findings.

Language and Style

While generally intelligible, the manuscript contains numerous grammatical issues and long, complex sentences. A thorough language review is recommended to enhance readability and flow.

AUTHORS:

We reduced sentence length, minimized passive constructions, and adopted more concise academic language throughout. The manuscript has undergone a full language and clarity review to enhance readability and flow.

REFEREE 5

The manuscript meets with the requirements with some adjustments and recommendations. Title: “Determining the Global Citizenship Tendencies of International University Students for a Sustainable World”

This manuscript explores an important and timely topic, how international students perceive and embody global citizenship, and how this relates to sustainable development. The mixed-methods design and sizeable sample strengthen the study. However, several issues related to theoretical framing, integration of findings, and data interpretation limit its contribution. Below are specific comments organized by theme.

Novelty and Contribution

The study topic is relevant, but the manuscript lacks a clearly defined research gap. The literature review broadly summarizes existing work but does not convincingly position this study as offering new insights. Therefore, it is suggested to clearly state how this study builds upon or differs from prior work, especially those using the Global Citizenship Scale or examining international students.

AUTHORS:

We revised the final paragraph of the Introduction to clearly define the gap in the literature.

We have added a paragraph at the end of the literature review section clarifying how this study extends prior work.

The differences with the 3 studies used in the literature are added in the conclusion section where the quantitative findings are summarized.

Theoretical Framework

The manuscript lacks critical engagement with existing theories of global citizenship. There is limited reference to alternative or critical perspectives (e.g., postcolonial critiques, tensions with national identity).

AUTHORS:

In the Discussion section, we now reflect more critically on the type of global citizenship being assessed (cosmopolitan/liberal orientation) and acknowledge limitations in framing.

Mixed-Methods Integration

While the convergent parallel design is appropriate, the manuscript does not sufficiently describe how the quantitative and qualitative data complem

---

## [Decision Letter · Decision Letter 1]

10 Sep 2025

Dear Dr. AKHAN,

Thank you for submitting your manuscript to PLOS ONE. After careful consideration, we feel that it has merit but does not fully meet PLOS ONE’s publication criteria as it currently stands. Therefore, we invite you to submit a revised version of the manuscript that addresses the points raised during the review process.

We look forward to receiving your revised manuscript.

Kind regards,

Abhik Ghosh

Academic Editor

PLOS ONE

Journal Requirements:

Reviewer's Responses to Questions

**Comments to the Author**

Reviewer #1: All comments have been addressed

Reviewer #3: All comments have been addressed

Reviewer #4: All comments have been addressed

Reviewer #5: All comments have been addressed

2. Is the manuscript technically sound, and do the data support the conclusions?

Reviewer #1: Yes

Reviewer #3: Yes

Reviewer #4: Yes

Reviewer #5: Yes

3. Has the statistical analysis been performed appropriately and rigorously?

Reviewer #1: Yes

Reviewer #3: Yes

Reviewer #4: Yes

Reviewer #5: Yes

4. Have the authors made all data underlying the findings in their manuscript fully available?

Reviewer #1: Yes

Reviewer #3: Yes

Reviewer #4: Yes

Reviewer #5: Yes

5. Is the manuscript presented in an intelligible fashion and written in standard English?

Reviewer #1: Yes

Reviewer #3: Yes

Reviewer #4: Yes

Reviewer #5: Yes

Reviewer #1: Overall, the manuscript is well-written and presents a clear and relevant discussion. The objectives are well addressed, and the arguments are supported with appropriate

references.

Reviewer #3: Abstract Revision Notes

1. Clarity of the problem/research gap

The abstract goes straight to the objective without mentioning why it matters or what the previous research gap is. International journals usually require an opening sentence or two that emphasizes the urgency/major topic and the research gap.

2. More concise methodological details

The methods section is somewhat lengthy. International abstracts typically condense: mention the design, sample size, and instruments, but avoid technical details like "using a protocol developed by the researchers." This is more appropriate for a methods article, not an abstract.

3. Consistency of results

There is little repetition between quantitative and qualitative results. These can be summarized to key insights and then emphasize the convergence, as this is a mixed methods study.

4. International academic language

Some phrases are still long and could be condensed for greater efficiency. For example:

a. Quantitative findings indicate a moderate level of global citizenship awareness among university students, could be condensed: Quantitative findings revealed a moderate level of global citizenship awareness.

b. Many participants reported applying global citizenship principles in their daily routines, Many participants reported applying global citizenship principles in daily practices.

5. Closing abstract

The final section mentions policy recommendations, but they are still generic. It should be more specific: for example, integrating global citizenship and sustainability education into the higher education curriculum is recommended to strengthen students’ role as agents of sustainable change.

So, the main things that need to be revised are:

1. Add context to the problem/gap at the beginning.

2. Summarize the methodology section (without technical details).

3. Condense and emphasize the novelty of the results.

4. Use shorter, more direct, and formal academic language.

5. Strengthen the takeaways in the closing section.

Keyword Revision Notes

keyword suggestions:

Global Citizenship; Sustainability; International Students; Higher Education; Education for Sustainable Development (ESD).

Introduction Revision Notes

1. Logical Flow Structure

The current introduction feels long and repetitive (for example, the sections explaining "global citizenship" and "sustainable development" are explained more than once).

2. Clarity of Research Gap

a. Currently, the introduction is mostly descriptive and normative (explaining the concepts of global citizenship and sustainability), but lacks clarity on the gaps in previous research it aims to address.

b. There should be a sentence like: "While previous studies have examined global citizenship awareness in different contexts [refs], limited attention has been given to international students in Turkey, particularly in relation to sustainability awareness."

3. Strong International References

Only a few citations appear at the end ([1,2]). International journals typically demand more citations from Scopus/Web of Science articles from the last 5–10 years, especially to support the claim: globalization, global citizenship, sustainability, higher education, international students.

4. Academic and Concise Language

a. Many sentences remain long and repetitive. For example:

b. “Global citizens value universal principles such as justice, human rights, cultural pluralism, and environmental awareness…” This could be condensed to “Global citizens value universal principles such as justice, human rights, cultural pluralism, and environmental awareness.”

c. Avoid repeating the phrases “global citizens” or “sustainable development” repeatedly.

5. The closing paragraph should be more explicit.

a. Currently, the research objective is stated, but it is still mixed with narrative.

b. It should be made explicit in one or two strong sentences: “Therefore, this study aims to determine the global citizenship tendencies of international students in Turkey and to explore how these tendencies are related to sustainability awareness.”

So, the most necessary revisions:

1. Condense the description of globalization and global citizenship (avoid repetition).

2. Add recent international citations to strengthen the framework.

3. Clearly identify the research gap (what has been studied and what is missing).

4. Refine the language to be more concise and formal.

5. Close the introduction with a clear statement of the research objective.

Method Revision Notes

1. The Method Structure Is Too Long and Mixed

The methods section is a mix of background, technical explanations of the instrument, procedures, and validity and reliability test results. This section should be separated.

2. Redundancy & Excessive Detail

a. For example, the description of the Global Citizenship Scale (22 items, factors, minimum-maximum scores, reliability, etc.) is too detailed. International articles should simply state who developed it, a brief structure, examples of dimensions, and reliability (Cronbach's α). Item details are better placed in an appendix.

b. The explanation of the interview procedures (schedule, WhatsApp, faculty location, etc.) is too technical. Generally, it should be sufficient: "Semi-structured interviews were conducted face-to-face, audio-recorded, and transcribed for analysis."

3. Clarity of Sample Selection

a. You've mentioned the number of participants, their region of origin, and inclusion criteria, good

b. However, information about the sampling method (purposive, convenience, random?) is needed.

c. In international journals, details on how to contact students via email/WhatsApp are too operational → better summarized.

4. Research Ethics

a. You've mentioned ethics approval (good!).

b. However, it's best to present it concisely: the ethics approval number, the licensing body, and a statement that informed consent was obtained. No need for technical details like "Google Forms asks 'Are you willing to...'".

5. Data Analysis

a. For quantitative research: too detailed explanations of normal distribution theory (skewness, kurtosis, Shapiro-Wilk test with p<0.05, etc.) can be summarized. Sufficient: “Normality assumptions were tested (skewness, kurtosis, and Shapiro-Wilk test), and as some variables were non-normally distributed, non-parametric analyses (Spearman correlation, Mann–Whitney U, Kruskal-Wallis) were conducted.”

b. For qualitative research: the explanation is good (MAXQDA, thematic analysis, constant comparison). However, it can be summarized without the step-by-step WhatsApp follow-up with participants.

6. Language & Focus

a. Some sentences are still “stories” rather than “methods reports.” For example: “Students were reminded to complete the scale sent via Google Form before coming for the interview.” Better written formally: “Participants completed the scale prior to the interview sessions.”

b. Avoid first person (“the researcher conducted…”); use the formal passive voice.

So, the main revisions needed are:

1. Divide the methods into standard subsections.

2. Concise the description of the measurement tools and procedures (leave the rest in the appendix).

3. Emphasize the type of mixed methods design and the rationale for its selection, but avoid overly narrative.

4. Concise the data analysis section (don't go into too much detail about statistical theory).

5. Strengthen the explanation of sampling (what type).

6. Make the language more formal and less "chronological."

limitations and implications Revision Notes

1. A clearer separation between limitations and recommendations.

Currently, limitations and recommendations are intertwined. For example, the sentence "Furthermore, although qualitative data reveals that students experience conflict between global and national identities, this important information cannot be examined in depth" is a limitation. But immediately after that, a recommendation is presented, making it difficult for the reader to distinguish between them.

2. Avoid repetition of ideas.

a. The idea "the need for students from diverse geographic regions" appears several times with different wordings. It could be condensed into one clear limitation.

b. Similarly, "conflict between global and national identities" is mentioned more than once. This could be summarized to make it more powerful.

3. Make limitations more concise and specific. Examples of revised wording:

a. Sample limitations (only students from certain regions, no representation of Europe, Australia, or developed Asian countries).

b. Data triangulation limitations (quantitative results cannot be confirmed with qualitative ones due to limited participants).

c. Issue exploration limitations (global vs. national identity conflict is not explored in more depth).

4. Make recommendations more structured.

a. Methodological recommendations: cross-national studies, involving students from more backgrounds, mixed-methods with a more balanced design.

b. Thematic recommendations: in-depth exploration of global versus national identity.

c. Practical recommendations: development of educational interventions, integration of global citizenship content into the curriculum, utilization of social media.

5. Avoid an overly normative or “long-winded” tone.

The section on “creating a more conscious global society in the long term” is more appropriate in the Discussion or Conclusion, not in the limitations/recommendations section.

6. More concise academic language. If you want an international level, use more concise wording, for example:

A major limitation of this study is the restricted diversity of qualitative participants, which excluded students from Europe, Australia, and developed Asian countries. Future research should address this gap through comparative cross-regional interviews to triangulate quantitative findings and deepen the exploration of identity conflicts between global and national citizenship.”

So the revisions are primarily:

1. separate the limitations and recommendations sections more clearly,

2. avoid repetition,

3. condense language,

4. focus on points directly relevant to the study.

Conclusion and Discussion Revision Notes

1. Mixed Discussion and Conclusions

a. Currently, the "Conclusions and Discussion" section is too long, with details like results and discussion.

b. Generally, the Discussion section is used to compare findings with the literature, provide interpretations, and explain implications.

c. The Conclusion should be concise, containing only answers to the research questions, main contributions, limitations, and suggestions for further research.

2. Excessive Repetition

Many points are repeated, for example, about moderate scores on global citizenship, the impact of social media, or regional differences. These points should be mentioned once to strengthen the argument.

3. Lack of Focus on Novelty

a. Currently, the findings are predominantly descriptive ("medium scores," "Europe is higher," "social media is influential"). It's important to emphasize what's new compared to previous research.

b. For example: is this study stronger because it compares multiple regions simultaneously? Or because it includes social media and global participation variables simultaneously?

4. Practical and theoretical implications haven't been separated.

Could you clarify: what contribution do these findings make to global citizenship theory? What are the implications for university or international institution policy?

5. The conclusion is too long

a. The conclusion should be brief (1–3 paragraphs), without further literature citations, and without repeating the results in detail.

b. Simply formulate the main findings, research contributions, limitations, and recommendations for further research.

6. International academic language

a. Some sentences are still long narratives. For international journals, it's best to condense them, using straightforward sentences, for example:

b. "The findings reveal that international students generally demonstrate a moderate level of global citizenship, with significant regional and experiential variations."

c. Then go straight to the interpretation "This suggests that while awareness exists, translating it into sustained responsibility and action remains limited."

So the revisions should be roughly:

1. Separate the Discussion (interpretation + comparison with the literature + implications) and the Conclusion (short answer, contributions, recommendations).

2. Concise the Conclusion section, removing detailed figures/variables already included in the results.

3. Clarify the novelty (what this study adds to the literature).

4. Add limitations and recommendations for further research (e.g. limitations of the Turkish context, predominance of respondents from certain regions, or limitations of the instrument).

Reviewer #4: I appreciate the authors’ thorough and constructive revisions in response to the previous review round. The manuscript has significantly improved in clarity, structure, and depth of discussion. The additional explanations regarding methodology, the intersectional framework, and the contextualization of findings have strengthened the paper.

The statistical analysis is now described with sufficient detail, and the presentation of results is more transparent. The authors have also addressed concerns regarding data availability and ethics, which are now satisfactory. The English usage is clear, and the text reads smoothly with only minor stylistic issues that can be handled during copyediting.

Overall, I find that the authors have adequately addressed the earlier concerns, and the manuscript is now technically sound, well-written, and ready for publication.

Recommendation: Accept.

Reviewer #5: Suitability for PLOS ONE: The article now aligns more closely with PLOS ONE’s requirements for methodological clarity, data availability, and policy relevance. Contribution: Moderate to strong. The study provides empirical insights into how international students in Turkey engage with global citizenship and sustainability, though theoretical framing could be richer. Recommendation: Accept with minor revisions Tighten the abstract and introduction. Add more nuance to the theoretical framework. Enrich qualitative results with additional quotes. Streamline results presentation for readability.

**Do you want your identity to be public for this peer review?** For information about this choice, including consent withdrawal, please see our Privacy Policy

Reviewer #1: No

Reviewer #3: No

Reviewer #4: **Yes: ** Alexander Adrian Saragi

Reviewer #5: **Yes: ** Uzlifatul Masruroh Isnawati

---

## [Author Response · Author response to Decision Letter 2]

13 Oct 2025

The recommendations made by the reviewers and the corresponding corrections are indicated in the response to reviewers file.

---

## [Editor Report · Decision Letter 2]

17 Oct 2025

Determining the Global Citizenship Tendencies of International University Students for a Sustainable World

PONE-D-25-07333R2

Dear Dr. AKHAN,

We’re pleased to inform you that your manuscript has been judged scientifically suitable for publication and will be formally accepted for publication once it meets all outstanding technical requirements.

Kind regards,

Abhik Ghosh

Academic Editor

PLOS ONE
---

## [Editor Report · Acceptance letter]

PONE-D-25-07333R2

PLOS ONE

Dear Dr. Akhan,

I'm pleased to inform you that your manuscript has been deemed suitable for publication in PLOS ONE. Congratulations! Your manuscript is now being handed over to our production team.

Kind regards,

on behalf of

Dr. Abhik Ghosh

Academic Editor

PLOS ONE